# Diverging land-use projections cause large variability in their impacts on ecosystems and related indicators for ecosystem services

Anita D. Bayer[1], Richard Fuchs[1], Reinhard Mey[2], Andreas Krause[3], Peter H. Verburg[4], Peter Anthoni[1], Almut Arneth[1]

[1]Karlsruhe Institute of Technology KIT, Institute of Meteorology and Climate Research, Atmospheric Environmental Research, 82467 Garmisch-Partenkirchen, Germany.

[2]Swiss Federal Institute for Forest, Snow and Landscape Research WSL, 8903 Birmensdorf, Switzerland.

[3]Technical University of Munich, TUM School of Life Sciences Weihenstephan, Hans-Carl-von-Carlowitz-Platz 2, 85354 Freising, Germany.

[4]Institute for Environmental Studies, VU University Amsterdam, de Boelelaan 1111, 1081HV Amsterdam, the Netherlands.

Correspondence to: Anita D. Bayer (anita.bayer@kit.edu)

## Abstract

Land-use models and Integrated Assessment Models provide scenarios of land use/cover (LULC) changes following pathways or storylines related to different socio-economic and environmental developments. The large diversity of available scenario projections leads to a recognizable variability in impacts on land ecosystems and the levels of services provided. We evaluated 16 projections of future LULC until 2040 that reflected different assumptions on socio-economic demands and modeling protocols. By using these LULC projections in a state of the art dynamic global vegetation model, we simulated their effect on selected ecosystem service indicators related to ecosystem productivity and carbon sequestration potential, agricultural production and the water cycle. We found that although a common trend for agricultural expansion exists across the scenarios, where and how particular LULC changes are realized differs widely across models and scenarios. They are linked to model-specific considerations of some demands over others and their respective translation into LULC changes and also reflect the simplified or missing representation of processes related to land dynamics or other influencing factors (e.g., trade, climate change). As a result, some scenarios show questionable and possibly unrealistic features in their LULC allocations, including highly regionalized LULC changes with rates of conversion that are contrary to or exceeding rates observed in the past. Across the diverging LULC projections we identified positive global trends of net primary productivity (+10.2% ± 1.4%), vegetation carbon (+9.2% ± 4.1%), crop production (+31.2% ± 12.2%) and water runoff (+9.3% ± 1.7%), and a negative trend of soil and litter carbon stocks (-0.5% ± 0.4%). The variability in ecosystem service indicators across scenarios was especially high for vegetation carbon stocks and crop production. Regionally, variability was highest in tropical forest regions, especially at current forest boundaries, because of intense and strongly diverging LULC change projections in combination with high vegetation productivity dampening or amplifying the effects of climatic change. Our results emphasize that information on future changes in ecosystem functioning and the related ecosystem service indicators should be seen in light of the variability originating from diverging projections of LULC. This is necessary to allow for adequate policy support towards sustainable transformations.

## 1. Introduction

The recently presented IPCC Special Report on Climate Change and Land (IPCC, 2019) highlighted unprecedented rates of land and freshwater use, biodiversity loss and underpinned existing socio-

economic, ecological and climatic challenges such as increasing per capita food consumption, land degradation and an accumulation of climate extreme events. The IPBES Global Assessment Report published earlier in 2019 (IPBES, 2019) also reported deteriorating levels of most ecosystem services (ES) and natural capital due to past and current human activities. The cumulative contribution of land-use and land-cover (LULC) change to global $CO_2$ emissions has been estimated to about one third of

total anthropogenic emissions since pre-industrial times (Friedlingstein et al., 2019), and total greenhouse gas emissions from LULC in recent years are nearly 25% of total anthropogenic emissions (IPCC, 2019). The diversity of current challenges towards a more sustainable use of land, including the maintenance of critical levels of resources and counteracting climate change, and the various options to approach these challenges create a large option space for possible future developments of LULC.

Future LULC and changes therein are modelled based on initial conditions of land use together with LULC history and different assumptions about possible socio-economic and environmental developments regarding population growth, international cooperation, consumption preferences or technological developments. All of these are represented differently in land-use models (LUM) or Integrated Assessment Models (IAM, e.g., DeFries et al., 2004; Meiyappan et al., 2014; van Vliet et al.,

2016). However, these models play a central role in assessing possible climate change mitigation and adaptation or conservation strategies in terms of total land demand, investment and maintenance costs, and direct and indirect socio-economic and ecological effects (e.g., Humpenöder et al., 2014; Popp et al., 2014; Reilly et al., 2012).

     In total, the diversity of models, initial model conditions, socio-economic pathways, climate mitigation

targets, processes and process feedbacks considered in the LULC modeling procedure leads to a large number of diverging land-use projections. This reflects not only the fact that the future is unknown but also a large uncertainty introduced by the model structure itself (e.g., Alexander et al., 2017; Prestele et al., 2016; Schmitz et al., 2014; Stehfest et al., 2019; van Vliet et al., 2016). By evaluating a large set of LULC projections (75 and 43, respectively), Alexander et al. (2017) and Prestele et al. (2016)

attributed a significant share of the uncertainty in global and regional LULC projections to the model initial conditions, resulting in part from different LULC definitions (especially for pastures, see also, e.g., Verburg et al., 2011), followed by the model structure, scenario storyline and other factors. Alexander et al. identified the differences in projected global LULC associated with the modeling approach to be at least as great as the differences due to scenario variations. In a regional-level

analysis, Prestele et al. found the highest uncertainty in land-use projections generally at the boundaries of boreal and tropical forests. LULC projections have also been evaluated in a number of model intercomparison studies, in which models simulated the same scenario storylines based on harmonized drivers, in order to focus on the uncertainty in LULC changes resulting from structural differences between the models (e.g., Von Lampe et al., 2014; Popp et al., 2017; Schmitz et al., 2014;

Stehfest et al., 2019).

     The large uncertainties in LULC projections affect the confidence in projected changes in ecosystem functioning globally, which critically underpins the supply of future ES available to human societies. In the same ways as the effects of climate change, the uncertainties arising from different LULC projections need to be identified and understood to adapt ecosystems in a sustainable way and

possibly counteract critical regional trends. Studies have focused on the vulnerability of ecosystems and their services to changes in climate (e.g., Ahlström et al., 2012; Huntingford et al., 2011; Ostberg et al., 2013; Scholze et al., 2006), land use on global or regional scale (e.g., Arora and Boer, 2010; Foley et al., 2005; Jantz et al., 2015; Krause et al., 2017; Lawler et al., 2014; Sterling et al., 2013) and a combination of climate and land-use effects (e.g., Dunford et al., 2015; Kim et al., 2018; Krause et al.,

2019; Rabin et al., 2020). Also uncertainties arising from different ES quantification methods were estimated (e.g., Schulp et al., 2014). These studies have already begun to document that diverging LULC projections are as important as diverging climate change scenarios in the degree of impact on ecosystems.

We therefore expand these previous studies here by bringing together a larger number of LULC
scenarios and by critically examining the resulting variability in diverging LUM projections based on
recent historical observations. We intend to also highlight how different LULC patterns impact
ecosystems and related ES indicators. This supports the interpretation of conclusions derived from
LUMs and IAMs towards policy decisions for instance on intensification, conservation or climate
change mitigation options. A broad range of future LULC projections from different LUMs and different
socio-economic assumptions is important, given the unknown future. Nevertheless, assessing critically
the spatial pattern and rates of change can support their interpretation in terms of plausibility.

Our basis were 16 projections of future land-use from five LUMs or IAMs with different modeling
protocols and socio-economic pathways. Their scenario storylines span a wide range of worldviews
and policies, with some implemented to achieve a certain climate mitigation or conservation target
while others focus only on basic demands for agricultural commodities, built-up area, etc. Models and
scenarios were assessed based on their underlying demands, modeling protocols (assumptions
involved, allocation strategies, etc.) and the projected spatially explicit land-use futures that they
describe. Then, we used the 16 land-use projections as input for simulations with a state of the art
dynamic global vegetation model (DGVM) to analyze their effects on ecosystem functionality and six
selected ES indicators linked to the productivity and carbon (C) sequestration potential of ecosystems
(net primary productivity, vegetation C, soil and litter C), agricultural production (crop production) and
the water cycle (evapotranspiration and annual runoff). We focused on changes until 2040, i.e. the
near to medium future.

## 2. Methods

### 2.1. Land-use models and scenarios

We used a total set of 16 land-use scenarios originating from five different LUMs or IAMs. The models
differ in their underlying demands, modeling protocols and technical aspects (e.g., number of
represented land-use classes, time horizons), which are summarized in this section and in Table 1.
Although some of the models considered here are IAMs including a land-use component, we refer to
all models in this study as global LUMs for the remainder of this paper because their projected LULC
change is the target of this analysis. This includes the two versions of the Land-use Harmonization
(LUH) project prominently applied in many studies of the last and the upcoming IPCC reports, although
these land-use products are based on the outputs of several LUMs/IAMs.

The CLUMondo model (van Asselen and Verburg, 2013) applies 30 land system types to model LULC
changes. Land systems define typical combinations of shares of cropland, grassland, bare land and
built-up land together with a specific management intensity (e.g., extensive cropland with few
livestock). Land systems are dynamically allocated based on local suitability, spatial restrictions and
the competition between land systems to fulfill demands that were created exogenously by the IMAGE
model on the level of world regions. Trade between world regions is excluded in CLUMondo. Eitelberg
et al. (2016) designed three CLUMondo scenarios: a reference scenario based on FAO-expected
developments of basic demands, and two scenarios that in addition included a policy target of reducing
deforestation and greenhouse gas emissions with a higher ecosystem carbon storage, and
international policy targets for the prevention of biodiversity loss.

The Integrated Model to Assess the Global Environment (IMAGE) is an IAM framework including sub-
models representing the energy system, agricultural economy, land-use and the climate system
(Stehfest et al., 2014). LULC allocation is done following an assessment and ranking of land's suitability
to fulfill demands. From IMAGE, a LULC baseline projection following increased food demand and
population growth according to the Shared Socioeconomic Pathway (SSP) 2 (O'Neill et al., 2014) is
available and two additional scenarios involving land-based climate-change mitigation, either via the

conservation and expansion of global forest area (ADAFF) or bioenergy crop cultivation and subsequent carbon capture and storage (BECCS) (Krause et al., 2017).

MAgPIE is a global land-use model of the agricultural sector (Lotze-Campen et al., 2008; Popp et al., 2014). It optimizes spatially-explicit land-use patterns in a recursive dynamic way to satisfy given commodity demands at minimal production costs while meeting biophysical and socio-economic constraints. Options to fulfill increasing demands are intensification (yield-increasing technologies), cropland and pasture expansion and international trade. Future land-use projections of the MAgPIE model follow the same storylines as described for IMAGE (Krause et al., 2017).

The Hurtt et al. (2011) modeling approach (LUH1) combines a historic land-use reconstruction with national statistics of historical wood harvest and assumptions regarding shifting cultivation in some tropical regions and harmonizes these data with a set of four future LULC scenarios. Each scenario was produced by a different IAM with each individual demands and strategies for allocating LULC in response to the demands. The four scenarios follow very different socio-economic storylines that are combined with the emissions and climate change assumptions of the Representative Concentration Pathways (RCPs). LUH1 scenarios are not tied to the SSPs as those were only introduced in 2014. These scenarios have frequently been used in the modeling community, especially for the work in the IPCC AR5 (e.g., O'Neill et al., 2014; van Vuuren et al., 2014). We used here the version of LUH1 which had the historical dataset extended until 2014 (Le Quéré et al., 2015), and future trajectories following IAM implementations of RCPs 2.6, 4.5, 6.0 and 8.5.

The Land Use Harmonization v2 (LUH2; v2.1, Hurtt et al., 2020) has been developed for the CMIP6 intercomparison project (Eyring et al., 2016). It follows a similar methodology as in LUH1, but on a higher spatial resolution and a longer time domain, using updated historical land-use reconstructions along with updated models of past and future land transitions and management (e.g., wood harvest, crop rotations and shifting cultivation) and extending the number of scenarios by combining RCPs with the SSPs. Similar to LUH1, future LULC transitions in LUH2 are based on land-use projections from different IAMs each following an own strategy for allocating LULC in response to demands. Of the eight scenarios that have been harmonized in LUH2 with historical data, three scenarios were selected (SSP1-26, SSP3-70, SSP5-85) to span the range from low to high radiative forcing as in the LUH1 scenarios in combination with diverging land-use trends according to the SSPs.

## 2.2. LPJ-GUESS model

The process-based dynamic global vegetation model LPJ-GUESS simulates vegetation dynamics in response to climate, atmospheric $CO_2$, land-use change (Lindeskog et al., 2013) and nitrogen (N) dynamics (Olin et al., 2015; Smith et al., 2014). Three distinct land-use types are represented (natural vegetation, pasture and cropland). Vegetation dynamics on natural areas are characterized by the establishment, competition and mortality of twelve plant functional types (PFTs, ten woody and C3 and C4 grass types, as in Smith et al., 2014), which are distinguished in terms of their bioclimatic preferences, photosynthetic pathways and growth strategies. Pastures are populated with competing C3 and C4 grass PFTs, where each year 50% of the above-ground biomass are removed as a representation of grazing (Lindeskog et al., 2013). Croplands are represented by prescribed fractions of crop functional types (CFTs, i.e. C3 crops with winter and spring sowing date, C4 crops and rice), with crop specific processes including dedicated carbon allocation and phenology, explicit sowing and harvest representation, irrigation, fertilization and unmanaged cover grass growing between cropping seasons (Olin et al., 2015). Crops are prescribed to be either rain-fed or irrigated (Lindeskog et al., 2013). LPJ-GUESS does not assume yield increases due to technological progress (such as advanced new varieties, management techniques, pest control), but yields respond to changes in climate, atmospheric $CO_2$ concentration, N input (deposition and fertilizer rates) and the fraction of rain-fed vs. irrigated cropland. Adaptation to climate change is partially accounted for by a dynamic calculation of

potential heat units (PHU) needed for the full development of a crop before harvest, simulating the
adequate selection of suitable crop varieties under changing climate (see Lindeskog et al., 2013). Upon
conversion of forested natural land for agriculture, 20% of the woody biomass enters a product pool
(turnover time of 25 years), with the rest being directly oxidized (74%) or decomposed as litter (6%).
Following agricultural abandonment, natural vegetation recolonizes the land in a typical succession
from herbaceous to woody plants, with competition for resources and light among age cohorts of
woody PFTs simulated directly through forest gap dynamics. In natural ecosystems, fire is simulated
explicitly as a recurring disturbance, while other episodic events (such as insect outbreaks or
windthrow) are subsumed in a background-disturbance occurring with a probability of 1% each year.

## 2.3. Simulation setup

LPJ-GUESS was run at 0.5° x 0.5° resolution forced by monthly climate of the IPSL-CM5A-LR general
circulation model (GCM). The model projects a global average surface temperature increase of
about 1.3°C by the end of the century relative to 1980-2009, which lies in the middle of an
ensemble of a wider range of GCMs used in the ISI-MIP intercomparison project (Warszawski et al.,
2014). Climate projections and $CO_2$ concentrations followed the RCP 2.6 pathway. Large magnitudes
of climate change and high atmospheric $CO_2$ concentrations affect ES indicators notably (see, e.g.,
Alexander et al., 2018). As our focus here is on the impact of land-use change, we chose a climate
change projection, which over the simulation period would have relatively little additional impact. In
a sensitivity experiment we explore the range of variability due to different climate models using the
RCP 2.6 outputs from GFDL-ESM2M, HadGEM2-ES, MIROC-ESM-CHEM and NorESM1-M models
(Warszawski et al., 2014) and LULC from the four LUH1 scenarios.

We modeled the ice-free land surface and included only those grid cells in our simulations for which
all LUMs provided data. Table S1 provides the detailed simulation set-up with all forcing data for the
LPJ-GUESS simulations. The differences in the modeling protocol of CLUMondo/LUH1/LUH2 and
IMAGE/MAgPIE simulations will affect the base level of ES indicators in 2000-2004 to some degree,
although the impacts of slightly diverging historical model periods, spin-up and historical climate would
have mostly disappeared by the beginning of the 21st century (baseline period). Larger effects arise
from the differences in the individual LUMs per se (see also Alexander et al., 2017). In principle,
differences in the baseline land-cover maps could spill-over to the simulated degree of change in the
future scenarios. For instance, presence or absence of natural vegetation in the baseline maps might
translate into variable degrees of future (semi)natural vegetation re-growth. However, this would only
be an important consideration when comparing similar scenarios (and their underpinning storylines
related to e.g. sustainability). The alternative approach of harmonizing the different projections to the
same starting point of land-cover would artificially mask some of the simulated differences in ES
indicators which would be contrary to our objectives. Therefore, LUM data were taken as they are,
with each LUM scenario providing a seamless transition from historical to future, which is needed to
simulate vegetation and carbon cycle responses.

Some of the variables assessed in LPJ-GUESS would also be computed in the models that deliver the
LULC change scenarios – most notably crop yields and some carbon-cycle or water-cycle related
variables. The spatial patterns of these would differ in the LUMs and LPJ-GUESS. However, this does
not affect our analysis: here we take the LULC change projections in a uni-directional approach to
assess impacts on ecosystem processes; we do not compare similar ecosystem output variables across
different model types.

LULC fractions were taken as net annual transitions from the LUMs and aggregated to the three land
use types cropland, pasture and natural land used by LPJ-GUESS (see Table S2) and to the spatial
resolution of 0.5° x 0.5° if needed. Cropland fractions also included bioenergy areas and pasture
fractions included degraded forests (IMAGE only), rangeland and grazing land. As LPJ-GUESS doesn't

represent urban land, built-up areas were included in the natural land fraction. Where 0.5° grid-cells contain substantial shares of water, this fraction was assigned to bare land, i.e. excluded from the simulation of vegetation pattern in LPJ-GUESS. In CLUMondo, LULC changes are occurring not between land cover types but between more complex land use systems (see section 2.1) that have different compositions in terms of natural, pasture and cropland area. In addition, the fractions of each land system vary regionally. In our simulations we did not include wood harvest. In the BECCS scenarios we assumed 80% of the harvested C from bioenergy crops to be captured and stored following Krause et al (2017).

### 2.4. Simulation of ecosystem service indicators

Changes in ecosystem function and services were assessed using a suite of regulating and provisioning ES indicators: net primary productivity (NPP, foremost an indicator for ecosystem productivity and C sequestration related to global climate regulation, also used as indicator for ecosystem health), C storage (natural capital that underpins and is closely related to C sequestration and global climate regulation), crop production (contributing to food supply), annual water runoff (indicator for water availability but also related to flood regulation) and evapotranspiration (indicator for regional climate regulation). C storage was investigated for vegetation, soil and litter C and total C with the latter also including carbon stored via CCS in BECCS scenarios. All plant and crop functional types contribute to an ecosystems' NPP. Therefore, NPP and crop production are positively correlated. All variables are direct outputs of LPJ-GUESS simulations. Baseline crop yields of each run were scaled to FAO observed yields in 1997-2003 as in Krause et al. (2017). Yields respond to changes in climate and $CO_2$, including also some degree of adaptation, which arises from the calculation of dynamic PHU (see 2.2). Adaptation related to, e.g., choosing different crop species in a grid-cell was not considered here in simulations of the future period. Changes in ES indicators were analyzed for each LULC scenario as percent change in 2036-2040 relative to the base level in 2000-2004. The average of 5 years was used to reduce the influence of inter-annual variability. Since climatic and atmospheric changes are identical across the simulations, differences in ES indicator changes across the scenarios reflect mostly the immediate and long-term effects of changes in LULC, also taking into account that climate change impacts might be dampened or amplified depending on the vegetation cover existing in a grid location. The evaluation of percent changes in future LULC and ES indicators relative to the baseline period in 2000-2004 partially takes account of differences in baseline LULC patterns and ES provision levels across the scenarios.

### 3. Results

### 3.1. Strategies of LUMs to translate demands into land use changes

Figs. 1 and 2 summarize LULC changes of the 16 projections. The scenarios projected total changes in LULC of about 4.5% to 11.4% of the global ice-free land surface from 2000 to 2040 (Table 2), corresponding to different transitions between crop, pasture and natural land. While different socio-economic assumptions realized as land-use change projections by the same model led only to small variations in the outcomes in terms of absolute rates (see Fig. 1, Table 2) and spatial patterns of LULC changes (see Fig. 2), the variation in LULC change between models was much more important, including for similar socio-economic scenarios. This highlights the importance of the differences in modeling strategies. In this regard, we realize that the outcomes as described are indicative for the model's behaviors for the particular scenarios considered in this study, which are not necessarily but very likely representative for the model's general behaviors upon projecting future LULC patterns. LUH1/2 are exceptions in regard to the changes in between scenarios because individual scenario data originate from different LUMs, therefore their data differ substantially between all scenarios.

In comparison to the other LUMs, CLUMondo shows rather small-scale LULC changes spread across large parts of the world (Fig. 2). The three scenarios have the same demands for livestock, crop production, etc. and the additional objectives in terms of C uptake and storage or biodiversity conservation did not introduce large variations. Therefore, differences between the CLUMondo scenarios are small (see Table 2). The biodiversity scenario leads to most land area changes due to land system classes that diverge to either system intensification in some regions or extensification in others. Demands are fulfilled by almost linear trends until 2040 based on the assumed scenario storyline and demand estimates (Fig. 1).

Land-use changes in the three IMAGE scenarios also affect most of the productive land areas globally. Compared to CLUMondo, spatial patterns are different and the spread of percentage area changes across scenarios was higher (globally 5-10% difference per LULC class for IMAGE scenarios, see Table 2). Global trends are not linear; some scenarios even reverse their historic trend (e.g., IMAGE_ADAFF scenario for pasture) or accelerate it (e.g., IMAGE_BECCS scenario for cropland), possibly driven by the introduction of new land-use policies. In IMAGE, food production meeting the underlying societal demand has large priority. The Base scenario accordingly increases pasture area at the cost of natural land (presumably to satisfy demand for animal products in the underlying "SSP 2" world) while cropland increases only slightly, presumably because yield increases satisfy the increasing food demand of a growing population. The afforestation and reforestation scenario IMAGE_ADAFF partially reverses IMAGE_Base by expanding natural land at the cost of pasture land, while IMAGE_BECCS in addition to pasture expansion as in IMAGE_Base also expands cropland areas at the cost of natural land. Spatially, the distribution of land-use classes among all scenarios differs little, indicating that demands from the Base scenario (e.g., population growth, diets, food demand, trade) outweigh specific scenario demands.

In MAgPIE, land changes only occur in specific regions or countries, but then massively (SE Argentina and Southern Brazil, some countries in Eastern Africa and parts of Southern and Eastern Asia), with the by far dominant change being cropland expansion. The three MAgPIE scenarios differ relatively little in time and space, only the afforestation and reforestation scenario shows some again very local natural area expansion. Trends over time are linear. Decisions of where land use change takes place to meet food and feed demand strongly depend on minimising the costs of land conversion. Here, some countries seem to provide substantially cheaper commodity prices than others, explaining the radical changes seen in the regions as listed above (compare also Fig. 2). Noteworthy, MAgPIE and IMAGE derive potential crop yields and ecosystem C densities from the same DGVM (LPJmL, Bondeau et al., 2007), even though internal yield scaling and forest growth curves are implemented differently. However, their spatial patterns are quite different, emphasizing the role of individual strategies to translate demands under similar biophysical constraints into LULC patterns. Also the land demand to meet the same CDR target was found to be larger in IMAGE than in MAgPIE (Krause et al., 2018).

In contrast, land changes in all LUH1 scenarios are large and occur in most of the productive land areas globally, reflecting both the highly diverging socio-economic storylines as well as their implementation by different IAMs (see Table 2). Trends over time are non-linear but involve multiple break points or gradual slopes. Interestingly, LUH1_26Be, which was developed by the IMAGE model (Hurtt et al., 2011), focusses on a broad expansion of croplands in tropical regions, while IMAGE_BECCS (however, most likely implementing a different degree of bioenergy growth), includes a massive re-location of pastures and also croplands to tropical and subtropical areas, respectively (Fig. 2). LUH1_45Aff and even more so LUH1_60Stab focus on massive expansion of natural areas in all global regions where forests can be sustained, while LUH1_85Pop expands pastures and secondarily croplands in tropical and subtropical areas. The attribution of specific spatial LULC patterns to model allocation strategies vs. scenario storylines is impossible for LUH1, and in the same way also for LUH2, because underlying IAMs and storylines differ between each scenario.

LUH2 scenarios also differ substantially, corresponding to the very different SSP storylines and RCPs combined with their origin from different IAMs. LUH2s SSP1-26 (also implemented by IMAGE but with different socio-economic assumptions than IMAGE_BECCS and LUH1_26Be), which includes options for both bioenergy crops and forest regrowth shows expansion of natural areas mostly in temperate (and some boreal) regions of the northern latitudes and also in Australia, some cropland expansion and a reduction in pastures (Figs. 1 & 2). LUH2_SSP3-70 in contrary results in a massive cropland expansion in some regions, in combination with a re-location of pastures, while LUH2_SSP5-85 (implemented by REMIND/MAgPIE) shows very large and concentrated regional dynamics, with cropland expansion similar to the MAgPIE scenarios as presented above.

In summary, most scenarios only agreed on a trend for cropland expansion at the cost of natural or pasture areas in terms of total area (see also Fig. 1). Moreover, the scenarios showed very diverse patterns on where and how these changes were realized. The deviation in LULC changes from 2000-2004 to 2036-2040 across the scenarios (Fig. S1) therefore showed major disagreement for cropland, pasture and natural areas. Standard deviations of changes in land area >20% across all scenarios were found for all three LULC classes over wide world regions, especially in SE South America, entire sub-Saharan and Eastern Africa and some regions in Europe and Southern Asia. This agreed in parts with features that were identified in earlier studies evaluating a set of multiple model LULC projections, such as in terms of global and regional trends (Schmitz et al., 2014) and the location of hotspots of uncertainty in LULC projections (Prestele et al., 2016). Given the diversity in socio-economic storylines and LUMs, these findings are not surprising, but highlight (1) the need to critically reflect on which of the observed LULC change patterns might be considered more or less realistic given historical regional developments in combination with environmental, economic and political constraints such as water availability, yield gaps and governance issues (see section 4.1), and (2) the need to explore the existing uncertainty in terms of future LULC regarding the implications for ES indicators beyond yields (see section 4.2).

**3.2. ES indicators for alternative LULC scenarios**

The 16 land-use scenarios resulted in very diverse levels of ES indicators in 2000-2004 and changes therein until 2036-2040 simulated with LPJ-GUESS (see Fig. 3 and Tab S3 for all results given in the following). Fig. 4 shows the spatial distribution of categories in ES indicator levels and their changes until 2036-2040, averaged across the 16 scenarios. We decided to also investigate averages to explore some overall emerging trends in ES indicators that result from the combined effects of climate and land-use change on ecosystem functionality. The average maps are complemented by the regional variability in ES indicators (right maps in Fig. 4, see also section 4.2) as a measure for the large between-scenario variability in ES indicators. Where regional variability is low, differences in LULC across scenarios are small and ES indicator changes can solely be attributed to climatic changes and/or changing $CO_2$ concentration together with the joint trend in LULC shown by all scenarios for this location.

The declining trend in natural areas (average decline of 0.9% ± 4.0% by 2036-2040 across 16 scenarios) as shown by most LUMs (Table 2) is balanced by the combined positive effect of increased atmospheric $CO_2$ concentrations, N deposition and warmer climate (especially in higher latitudes) leading overall to an increased global vegetation productivity (+10.2% ± 1.4%) and higher total C stocks (+1.4% ± 1.1%) across the scenarios. Simulated changes agreed in the trend but levels were below those reported in previous studies (compare LPJ-GUESS simulations including LULC changes for IPSL-CM5A-LR climate of Brovkin et al., 2013; Pugh et al., 2018), noting that these studies applied different LULC data and used LPJ-GUESS without C-N limitation and with differing model set-up. Regionally, increases in vegetation productivity and C stocks were pronounced in boreal and temperate forests, while in the tropics, positive effects of especially $CO_2$ fertilization and improved water use efficiency (see, e.g., Wårlind et al., 2014) were reduced by cropland and pasture expansion, jointly with negative effects of warmer

and drier climate. Across the 16 scenarios, the increase in NPP and C storage was generally higher in CLUMondo, LUH1 and LUH2 than in IMAGE and MAgPIE scenarios. Increases in vegetation and total C stocks were, as expected, large in scenarios that showed significant amounts of forest regrowth (especially LUH1_45Aff, LUH1_60Stab, LUH2_SSP1-26, but also IMAGE/MAgPIE_ADAFF) and low in scenarios with agricultural expansion for food (e.g., IMAGE_Base, LUH1_85Pop) or bioenergy

production (IMAGE/MAgPIE_BECCS, LUH1_26Be). The overall changes in total C stocks reflected an increase in vegetation C (+9.2% ± 4.1%) that was balanced to some degree by a decrease in soil and litter C stocks (-0.5% ± 0.4%), likely driven by enhanced respiration of organic material under warmer temperatures (see, e.g., Pugh et al., 2015), in combination with the negative effects of decreasing natural areas on soil and litter C in most scenarios. The simulated increase in vegetation C was

significantly lower and the decrease in soil and litter C larger for all IMAGE and MAgPIE scenarios because in these scenarios, the conversion of natural land to pastures (for IMAGE) and to croplands (for MAgPIE) was largest among the analyzed scenarios.

Crop production was simulated to increase on average across all 16 scenarios by 31.2% ± 12.2% partly as a result of total cropland area increasing (+11.7% ± 10.5%, Tables 2 & S3) for all scenarios except in

LUH1_45Aff, and partly due to increasing yields. Yield increases resulted from the joint effects of increased N fertilization rates, warmer temperatures in some regions and increasing atmospheric $CO_2$ (see Fig. S4 in Krause et al., 2017). Crop production increases were found in all world regions, especially southern and eastern Asia, central and southern Africa, SE South America and cropping regions in North America and Europe. Differences in crop production between scenarios were due to different

absolute area and the location of cropland expansion on the globe (and differences in N fertilization rates for IMAGE and MAgPIE scenarios, see methods). For LUH1_45Aff, the simulated global total increase in crop production was only 2.6% because of the immense amounts of natural area expansion in this scenario reducing total cropland area in contrast to the other 15 scenarios. Furthermore, LUH1_60Stab and all IMAGE scenarios showed lower increases in crop production than the other

scenarios due to only small cropland expansion (in case of LUH1_60Stab) and newly established croplands being chiefly located in low to medium production areas (in case of IMAGE scenarios, e.g., Sub-Saharan and northern Africa, Middle East). For all LUH2 scenarios, crop production increases were high with about 44% relative to the level in 2000-2004. Lower crop production was simulated in IMAGE and MAgPIE climate change mitigation scenarios in comparison to their baseline scenarios and also in

CLUMondo scenarios when additional land-demands had to be met in comparison to their FAO reference scenario. This highlights the inherent trade-off created through multiple demands. It has to be noted that IMAGE (and therefore also CLUMondo because its demands are created by IMAGE) and MAgPIE internally calculate further technology applications (for example improved management, enhanced fertilizer inputs, pest control and better crop varieties) to increase yields in mitigation

scenarios up to the level of their baseline simulation. However, these are not fully captured by LPJ-GUESS (see methods).

Annual water runoff was simulated to increase on average by 9.3% ± 1.7% until 2036-2040 (ranges and average trend are similar to estimates from Elliott et al., 2013, based on ten global hydrological models). The increase resulted from the combined effect of increasing global total precipitation (+5.1%

from 2000-2004 to 2036-2040 in the IPSL-CM5A-LR model), the increased water use efficiency under elevated $CO_2$ levels (see, e.g., De Kauwe et al., 2013; Qiao et al., 2010) and changes in the water use of agricultural vs. forested areas that were shown in many studies (such as reduced evapotranspiration of croplands in comparison to forests, see, e.g., Farley et al., 2005; Sterling et al., 2013). Also changes in irrigation patterns affect water runoff (IMAGE and MAgPIE simulations only, see methods). All these

effects are captured by LPJ-GUESS (e.g., Krause et al., 2017; Rabin et al., 2020). Increases in runoff were simulated in the temperate zone and higher northern latitudes and in smaller regions in the tropical and subtropical zone. In water limited regions such as the subtropics, some of this water could in principle be available for irrigation, depending greatly on the regional annual runoff dynamics. But it will also increase erosion of soil and nutrients (e.g., Salvati et al., 2014) and the risk for floods in some

regions (see, e.g., Rabin et al., 2020), likely also intensifying regional dependencies of water availability and usage that are discussed elsewhere (see, e.g., Elliott et al., 2013; Fitton et al., 2019). Differences in runoff levels in 2000-2004 and changes until 2036-2040 were small between the 16 scenarios because the forcing climate dominates the calculated water balance, rather than LULC changes. Only for the three LUH2 scenarios, about 5% lower absolute levels compared to the other scenarios were
simulated in 2000-2004 and relative increases in runoff were about 3% larger for IMAGE and MAgPIE scenarios than the other scenarios.

Changes in evapotranspiration are closely linked to the calculations of runoff, although their effects are opposed, with higher evapotranspiration rates contributing to reduced surface runoff (e.g., Piao et al., 2007), but also to biophysical cooling (e.g., Anderson et al., 2011) Evapotranspiration rates
increased in CLUMondo, LUH1 and LUH2 scenarios on average by 2.6% ± 0.3% and decreased in the IMAGE and MAgPIE scenarios on average by -0.7% ± 0.5%. Increases in evapotranspiration rates in non-tropical regions likely reflect the expansion of forests (see, e.g., Sterling et al., 2013) and therefore were highest in the scenarios assuming intensive expansion of natural areas (LUH1_45Aff, LUH1_60Stab). This trend was balanced in the three IMAGE scenarios by the effects of large-area
conversion of forests to pastures in tropical South America and Africa leading to a total reduction in evapotranspiration rates. BECCS activities seem to reinforce the reduction in evapotranspiration rates, while the extension of natural areas counteracts it to about 50%. In MAgPIE scenarios, a strong increase in cropland area in SE South America and Eastern Africa corresponded with a small overall decrease in evapotranspiration.


## 4. Discussion

### 4.1.    Projected global LULC patterns in a historical context

**(1) Global change rates.** The 16 scenarios project total changes in LULC of between 4.5% and 11.4% of the global ice-free land surface from 2000 to 2040. These rates are of a magnitude comparable to those
observed for the past. For instance, four historical LULC reconstructions for the period 1960-2000 estimated changes in LULC of 7.5%-12.8% (Hurtt et al., 2011; Klein Goldewijk, 2016; Klein Goldewijk et al., 2011; Ramankutty et al., 2008, see Table S4). A more recent global historical LULC change reconstruction by Winkler et al. (in prep.) estimated net LULC changes of 13.8% for 1960-2015. This reconstruction uses a data-driven approach that is strongly based on remotely sensed information and
provides higher spatial, temporal and thematic resolution than previous reconstructions. For a shorter time-period, Liu et al. (2018) based on the ESA CCI Land cover product identified 3.4% net LULC changes for 1992-2015. The picture becomes more complex when gross transitions, rather than net transitions, are considered, since gross changes (e.g., from forest to cropland in parts of a grid location and cropland to forest in another, over one time-step) can be substantially larger than the net change. For
instance, change rates of Winkler et al. are 36.4% when gross and multiple LULC changes are being considered individually. By contrast LUM projections have simplified representation of these more realistic LULC dynamics, omitting short-term, two-directional or small-scale transitions (e.g., such as under shifting cultivation, e.g., Heinimann et al., 2017). The improved representation of gross land-use changes in historical and future LULC reconstructions would be an important development to better
account for LULC dynamics and their impacts on ecosystems (e.g., Bayer et al., 2017). However, such efforts are still hampered by a limited process understanding and data availability at the global scale.

**(2) Future regional change rates in a historical context.** Even given different initial states in LULC and different socio-economic pathways of the 16 scenarios, we critically assess the spatial patterns, directions and rates of regional change based on past LULC changes. While it is not completely
impossible, of course, we argue that a speed and magnitude which extremely oppose trends observed in the past seem at least questionable. Scenarios projecting future demands under reference or business-as-usual assumptions (CLUMondo_FAOref, IMAGE/MAgPIE_Base) are expected to continue

recent historic LULC trends at least during the first part of the simulation period. Nevertheless, economic growth assumptions, demographic considerations and limitations in land availability in the scenarios will likely cause some divergence from historical LULC trends. Historic trends are indeed continued in some of the broader regional projected trends, such as the expansion of natural areas in mid to high northern latitudes (CLUMondo_FAOref, IMAGE_Base, see Fig. 2) or the expansion of croplands in subtropical areas (CLUMondo_FAOref, to some degree also IMAGE/MAgPIE_Base) (e.g., Hansen, 2013). In contrast, some projected LULC changes have no historic precedent, such as the extensive pasture increase in tropical Africa (IMAGE_BASE), the large cropland expansion in some selected African countries (MAgPIE_Base), or cropland expansion in Mediterranean and Middle East (IMAGE_Base).

Scenarios including more drastic changes in the socio-economic system (e.g., all LUH1 and LUH2 scenarios), including specific conservation measures or large-scale land-based climate change mitigation efforts (e.g., CLUMondo_CStor/Bdiv, IMAGE/MAgPIE_ADAFF/BECCS), affect future LULC patterns in very different ways, compared to reference or business-as-usual scenarios. Whether or not simulated abrupt LULC changes or a rapid reversion of regional historic LULC trends are realistic is difficult to judge, as analogous historical evidence is scarce or absent. Indeed, some rapid land-use changes have occurred in the past, caused by unexpected disruptions in markets or governance structures (e.g., Brazil's soy moratorium combined with enforcement of related policies, e.g., Nepstad et al., 2014; Gibbs et al., 2015; collapse of the Soviet union, e.g. Hostert et al., 2011). However, capturing such unexpected LULC changes in global LUM projections is nearly impossible. Still, in response to most policy interventions, work has suggested that transitions in land use across regions tend to occur rather smoothly and with time lags of years to few decades (i.e. spanning a notable part of our simulation period) due to delayed policy uptake (e.g., Brown et al., 2019). In this context, large-area, and relatively rapid regional change rates could be assessed critically, such as (1) forest regrowth on pasture and cropland areas with more than 40% area change from 2000-2004 to 2036-2040 in SE South America (LUH1_45Aff) or entire subtropical Africa (LUH1_60Stab), (2) massive cropland increases exceeding 40% total area change, e.g., in SE South America and eastern Africa (MAgPIE_ADAFF/BECCS and LUH2_SSP5-85), and (3) pasture expansion exceeding 20% of the total area in tropical and subtropical Africa (e.g., IMAGE_BECCS, LUH2_SSP3-70). The first example reverses current deforestation trends (compare, e.g., Curtis et al., 2018; Hansen et al., 2013), while in the latter two examples, simulated rates of change are substantially larger than observed in these regions in recent decades (e.g., Hansen et al., 2013; Klein Goldewijk, 2016, compare Fig. S3). LULC scenarios assuming significant amounts of forest regrowth, should also be seen in light of the recent evidence provided by Holl and Brancalion (2020), pointing out manifold problems attached to tree planting and therefore calling for the prioritization of natural forest protection.

**(3) Regional LULC allocations.** In general, CLUMondo and IMAGE were more capable of capturing small-scale changes within heterogeneous regions. Given the complexity in which land changes are being observed (e.g., Curtis et al., 2018; Hansen et al., 2013), the capacity to simulate changes at small scale is likely more realistic. For instance, Stepanov et al. (2020) found for a case study in Brazil that a spatially explicit regionally LUM creating small-scale changes simulated observed LULC patterns much better than the larger-scale changes of an economic model. By contrast, regional patterns in other LUMs tended to be fairly broad in extent or were limited to regions that were small in area. This is not consistent with what can be learned from global multi-temporal remote sensing.

When assessing the plausibility of regional LULC allocations, a good indicator is the expansion of agricultural areas (e.g., Salmon et al., 2015) based on existing yield potential (i.e. fertile areas currently not used or with a low share of croplands) or based on existing yield gaps (i.e. because of poor

management or limits posed by the socio-economic environment). We find that the LUMs use the regions with existing yield capacities in different ways for their allocation of LULC types. Cropland areas with currently relatively large yield gaps, such as in Brazil's Cerrado, west or east Africa (e.g., Mueller et al., 2012), were used by CLUMondo for further cropland expansion, which seems plausible also considering past LULC trends and continuing economic growth in response to an increasing population in these regions. In IMAGE simulations (especially IMAGE_Base/BECCS), these regions were typically converted into pastures and instead croplands in IMAGE were expanded in northern Africa and western Asia (especially Syria and Iraq). Besides current political turmoil in north Africa and west Asia, these allocations seem less plausible considering rather unfertile soils and existing yield gaps that are predominantly linked to low fertilizer inputs and water scarcity (e.g., Pala et al., 2011). In addition, the expansion of cropland area in IMAGE in northern Africa (especially Libya and Egypt, see Fig. 2) notably exceeds current cropland extent (e.g., Fritz et al., 2015). It seems doubtful whether this can be indeed achieved, given that existing yield potential in these regions is low due to biophysical limitations during the crop-growing season which is not projected to change in the future. Other examples where LUMs diverge notably include central India, where all IMAGE scenarios used areas with large yield gaps as indicated by Mueller et al. (2012) for cropland expansion. In contrast, all CLUMondo scenarios used these croplands for forest regrowth, thus ignoring their potential to contribute to fulfill increasing food demand. The massive, but very regional, cropland expansion of all MAgPIE scenarios and LUH2_SSP5-85 includes regions in eastern Africa, where yield gaps are high, but also regions in SE South America that are already under cropland usage to a high degree and attained yields are high (e.g., Fritz et al., 2015; Mueller et al., 2012). This suggests that in the MAgPIE model, economic considerations dominate the allocation of LULC classes rather than existing biophysical capacities. Only few scenarios (e.g., LUH1_26Be, LUH2_SSP1-26, LUH2_SSP5-85) expand cropland area in continental eastern Europe, especially along the "Chernozem-Belt" into Russia, where soils are fertile and closing yield gaps would be expected to lead to large returns in terms of enhanced productivity (see Mueller et al., 2012). Other scenarios saw no potential for cropland expansion in these regions and simulated forest regrowth.

It has to be noted that in regions where attained yields are relatively close to potential yields already now (e.g., NE North America, W Europe, some parts of S and SE Asia, see Mueller et al., 2012), yields may decline in the future due to climate change (e.g., Elliott et al., 2013; Funk and Brown, 2009; Lobell et al., 2009; Moore and Lobell, 2015; Pugh et al., 2016), and unless this can be counteracted by different management (e.g., increased irrigation) or different crop varieties, production may need to shift to other regions. However, the degree of climate change over our simulation period is too small to discern negative impacts on yields, and associated climate-driven crop area changes.

Aside of biophysical considerations, LULC changes in response to changing economic conditions as projected in the scenarios require support by adequate regional policy, production and trading systems as well as appropriate technological capabilities (e.g., Lambin et al., 2003; Meyfroidt et al., 2019). Weak governance structures, for instance, would allow highly market-driven LULC changes, especially arising from changing demand elsewhere. LUM projections involving large-scale separation of LULC types across global regions (i.e. large regions of just one LULC class, e.g., IMAGE_Base/BECCS, all MAgPIE scenarios, LUH1_60Stab, LUH2_SSP5-85) could be interpreted to be detrimental to regional to national food production systems. This would include, for instance, subsistence farming systems over wide parts of Africa, which are an essential pillar of Africa's food supply and will continue to be such in the future (e.g., Sulser et al., 2015). In addition, technological capacities would have to be in place to support crop production at the projected locations possibly including irrigation, the use of appropriate machinery, fertilization, etc. (see, e.g., Barrett and Toman, 2010; Lambin et al., 2014; Nilsson and Persson, 2012; Wang et al., 2016). At present, no global LUM is set-up to consider governance aspects, such as land tenure rights or location-specific management, or transportation and trade capacities (apart from assumptions made in the economic core of LUMs that apply to large regions), which is an important need for further development.

To better understand regional LULC allocations and the related impacts on ecosystems and ES, the availability of more spatial information from LUMs related to regional-scale assumptions on technological progress, flows of food import, export and local production would be useful. In this context it seems worthwhile for the land use community to evaluate simulated future land-use changes against historic trends, in spatial, temporal and thematic aspects. This may avoid some of the questionable, possibly unrealistic, land-use change effects seen in this study. However, current data products of historic land-use change are often themselves associated with high uncertainty in historic trends, due to data limitations. Improved historical products that merge multiple data sources could support the evaluation of future projected LULC changes. In addition, a clear declaration of scenarios showing potentially possible regional or global LULC pathways in comparison to those showing LULC changes going beyond historical exemplars would be desirable.

**(4) Impacts on ecosystems and ES indicators.** It is well known that different climate trajectories (e.g., for different RCPs) will greatly affect ecosystems. Even climate change projections for a single RCP when realised with different ESMs will result in large variability in computed ecosystem outcomes (e.g., Ahlström et al., 2012). The fact that similarly large variability can be introduced by land-use change (within or between e.g. an SSP) is less established and as such an important outcome of this study.

The LULC patterns observed in the 16 scenarios suggested a general prioritization of food and feed demand affecting croplands and pastures (production of crops, meat, etc.) over those related to natural land dynamics (e.g., C storage, biodiversity). This is not surprising, given that aspects that could impact "non-food" demands, such as C prices are not considered in many of the scenarios' underlying storylines. In LUMs where both, a baseline scenario and scenarios with additional demands, were simulated (CLUMondo, IMAGE, MAgPIE), the overall trends in LULC and ES indicator changes were chiefly determined by the demands of the baseline scenarios and their model-specific implementation. Specific additional demands, e.g. for land-based mitigation or conservation, mostly resulted in only small deviations of LULC patterns and ES changes from the baseline scenarios. Deviations between scenarios of different models were much larger.

Scenarios that included specific climate change mitigation targets (CLUMondo_CStor, IMAGE_ADAFF/BECCS, MAgPIE_ADAFF/BECCS) resulted in larger total C storage in LPJ-GUESS, but lower crop production, compared to the baseline scenarios. In LUMs, the mitigation scenarios include technological-driven yield increases, which are higher than those assumed in LPJ-GUESS (see methods). From a C storage perspective, concern about the C storage potential calculated by LUMs was raised by Krause et al. (2018), who could not reproduce the cumulative C uptake that was calculated in IMAGE and MAgPIE BECCS and ADAFF when applying their LULC change to ecosystem models. On average only 62% of C uptake was achieved with LPJ-GUESS, and about 55%, when three other DGVMs were used in addition. Likewise, Harper et al. (2018) also found the IMAGE C storage potential from BECCS to be achieved by less than 25% upon simulating two IMAGE mitigation scenarios with the JULES DGVM. These discrepancies likely arise from different assumptions in LUMs and DGVMs related to growth rates and C uptake of re-growing forests and bioenergy crops, changes in soil C stocks upon LULC change, legacy effects of previous land-use changes and some further processes such as disturbances, e.g., forest fires. By contrast, when using the CLUMondo LULC patterns, all three scenarios led to an increase in total C storage in LPJ-GUESS, thus even exceeding the no-net carbon loss target that was implemented in CLUMondo_CStor. This might be explained by the joint effects of N deposition, $CO_2$ fertilization and climatic change that are core components of the LPJ-GUESS model but which were not implemented in similar detail in the CLUMondo calculations.

A number of studies have begun to identify in more detail how different assumptions in LUMs might affect LULC projections. Stehfest et al. (2019) recently provided a comprehensive sensitivity analysis of the socio-economic drivers that were projected across five SSP-based storylines by six agro-economic models/IAMs and found very diverging sensitivities across models. The study highlights the existing variability in LULC modelling and emphasizes the need for more empirical research on crucial

factors in the LULC modelling process such as long-term drivers of LULC change or the representation of land-use regulation and trade. The spread between LULC projections of different models could also be reduced by joint calibration and validation standards, but these are currently not existing (e.g., van Vliet et al., 2016). Reducing the spread between models, especially for similar scenario assumptions, would provide an important step forward in understanding LUMs LULC patterns and in identifying possibly implausible allocations and would also support the assessment of calculated impacts on ecosystem functioning and ES. A comprehensive comparison of the socio-economic drivers of LUMs with the ES indicator levels simulated with DGVMs based on the LULC patterns from LUMs would be needed. This would provide deeper insights on the dependencies between drivers, modeling strategies and resulting ES provisions and would contribute to identifying the quality of the representation of interactions between socioeconomic and environmental systems including relevant feedback mechanisms.

## 4.2. Variability in the future of global ES indicators

Despite the partially very diverging LULC projections, the 16 land use scenarios on the level of global totals (see Fig. 3, Tables 2 & S3) resulted in a positive change in NPP, vegetation C (all except one scenario), crop production and annual water runoff, and in a negative change of soil and litter C stocks (all except one scenario) from 2000-2004 until 2036-2040 when simulated with LPJ-GUESS. Diverging trends were predicted for evapotranspiration. Emerging trends are the result of joint climate change, increasing $CO_2$ levels (even under RCP 2.6) and current as well as past LULC changes on ecosystem functionality. We didn't quantify relative impacts of these factors separately. A sensitivity test using climate inputs from five GCMs, instead of just one, along with the four diverse scenarios from the LUH1 product in our simulation set-up showed additional uncertainties between 0.1% and 8.6% for the ES indicators considered here (Table S6). The lowest deviation due to different GCM implementations was found for crop production and the highest for vegetation C stocks.

Regionally, large variability in ES indicators (see right maps in Fig. 4 and Fig. S2) reflected the diverging LULC scenarios (effects from scenario storylines and model-specific implementation) and their interactions with climatic changes. Only in areas colored white, the absence of variability in changes of ES indicators indicate dominating climate change and $CO_2$ impacts. Variability in the predicted changes of global totals in ES indicators exceeded a low level of 1-2% for vegetation C stocks (± 4.1%) and crop production (± 12.2%) with regional variabilities (standard deviation of relative changes across scenarios) being high for these two indicators in nearly any productive region (Fig. 4). We discuss the observed regional variability in ES indicators on the level of biomes as large regions with similar ecological constraints (see Fig. S4 for biome classification).

Tropical forest regions, or at least major regions therein, were identified as hotspots of variability across the LULC projections with large areas showing variability >10% change across the scenarios for all ES indicators considered in this study (see Figs. 4 & S2 and Table S7 for biome averages of ES indicators). This is a result from the high vegetation productivity, large biomass and a relatively higher $CO_2$ fertilization on the one hand and very diverging trends in the LULC changes across the 16 scenarios, especially at the borders of the currently forested tropics (see also Prestele et al., 2016), on the other hand. For instance, losses of soil and litter C stocks are the net effect of higher decomposition rates as a consequence of a warmer climate in combination with higher inputs under $CO_2$-driven increased productivity. These climatic effects are strongly reinforced by the conversion of forests to croplands (LUH1_26Be) and pastures (all IMAGE scenarios, LUH2_SSP3-70, LUH2_SSP5-85, to some degree also CLUMondo scenarios) leading to even lower soil and litter C stocks. At the same time, they would be

attenuated through forest regrowth (LUH1_45Aff, LUH1_60Stab, MAgPIE_ADAFF), reducing regional carbon losses or even resulting in gains in soil and litter C stocks. As is well documented, future increased use and fragmentation of tropical forest ecosystems can be a major threat to conserving

tropical ecosystems and biodiversity (e.g., DeFries et al., 2005; Lewis et al., 2015; Taubert et al., 2018) and the existing protected area network is insufficient to provide the necessary protection (Laurance et al., 2012). Developing joint biodiversity and carbon storage policies might lead to possible reinforcing synergies (e.g., Strassburg et al., 2019), although appropriate governance schemes to support such attempts would still be a crucial factor.

In tropical savannas and temperate shrubland and grassland regions, the diverging LULC projections resulted in high variabilities for vegetation productivity, vegetation C storage, annual water runoff and crop production. Although 15 out of 16 land-use scenarios increase cropland area in these regions, they diverged widely in the exact location of cropland expansion resulting in these high variabilities in ES. Savannas have been highlighted before as particularly vulnerable to future conversions of natural

vegetation into cropland or pasture (e.g., Shin et al., 2019) because of large population growth in many savanna regions, their climatic suitability for agriculture and relatively small efforts needed for conversion considering the relatively low woody vegetation cover. Parts of these regions experienced intense cropland expansion already in recent decades, such as South America's Cerrado and Chaco or African savannas (e.g., Aleman et al., 2016; Hansen et al., 2013; Noojipady et al., 2017). Processes

affecting ecosystem functionality and services in these subtropical to semi-arid regions are particularly important also in view of rapid population growth and associated demands (Alexandratos and Bruinsma, 2012) and also their role in global C cycle and climate dynamics (e.g., dominant role for the trend and interannual variability of the global land C sink, see Ahlström et al., 2015, with their semi-arid class widely corresponding to our classes tropical savannas and temperate shrublands and

grasslands). Therefore, high variability in the provision of ES indicators related to vegetation productivity, water availability and food supply may have severe consequences on ecological, economic and social systems in these regions.

In both temperate and boreal forest regions, the regional variability of vegetation productivity and vegetation C storage was high because the increased photosynthetic productivity and longer growing

seasons under warmer climates in higher latitudes were either dampened or amplified by diverging LULC projections. While LULC changes dominate this combined positive effect in temperate regions, they are much smaller in boreal regions and the climatic effect dominates (see Table S5, compare also high variability of changes in NPP, vegetation and soil/litter C in boreal regions for multiple DGVMs, GMCs and emission pathways in Nishina et al., 2015, under fixed LULC). Counteracting these positive

trends, studies highlight new challenges for temperate and boreal forests emerging from climatic changes and anthropogenic disturbance with the capacity for severe ecosystem-level damages, such as droughts, insects, fire regimes and pathogens (e.g., de Groot et al., 2012; Millar and Stephenson, 2015; Park et al., 2014). As only fire is explicitly simulated as a disturbance process in LPJ-GUESS, while other forms of disturbances are subsumed in a stochastic background-disturbance, this could further

increase the regional variability in ES indicators(see, e.g., Pugh et al., 2019). Regional variability was high also for crop production due to diverging extensification and intensification trends across temperate and boreal regions. Land-use changes and associated regional variability in ES indicators in the cold (tundra) and warm desert regions are not significant in a global context because of the climatically constrained low productivity, although it is enhanced at least in tundra regions through

warmer temperatures.

A direct correlation of the per grid-cell changes in ES indicators with the corresponding changes in cropland, pasture and natural land fraction could reveal the sensitivity of different ES indicators to changes in LULC. Across all scenarios and for the considered biomes, these relationships suggested for instance an about 1.5% increase in vegetation C per percent increase in natural land fraction and between 12 and 24% increase in crop production per percent increase in the cropland fraction (see Table S8). Emergent responses of other ES indicators to changes in LULC fractions were mostly low (slope of regression lines close to 0) across the biomes (see Table S8). None of the identified relationships provided high reliability (highest $R^2$ was 0.32 for the change in vegetation C per change in natural land fraction in tropical forests, see Fig. S5). This reflects that direct correlations of ES indicator changes with changes in LULC are difficult to establish because they are significantly impaired by, e.g., overlying climate effects, different base levels in ES indicators and LULC configurations and different ecosystem responses below the level of biomes (including, e.g., legacy effects of past LULC changes).

**4.3. Significance of our approach**

The IPBES report on plausible futures of nature identified some general trends across their scenarios (e.g., continued increase in managed land, increases in material and decreases in regulating and non-material nature's contributions to people) and rated the knowledge base and confidence in effects of interactions of future LULC and climatic changes on biodiversity and ecosystem functioning with "established but incomplete" (see Shin et al., 2019). By evaluating 16 scenarios of five structurally different LUMs, we covered a large variability existing in currently available spatially-explicit projections of LULC and therefore contributed to extending the existing knowledge base on variability of future ES indicators. However, conclusions drawn here in regard to projected changes in LULC and ES indicators are inherently dependent on the selected set of LUMs and scenarios, evaluation time period and simulation set-up. In addition, we did not consider climatic variability resulting from different GCM implementations of climate under RCP 2.6 nor other possible emission pathways, which both adds uncertainty to future ES indicator levels. Previous studies (e.g., Friend et al., 2014; Krause et al., 2019; Nishina et al., 2015; Pugh et al., 2018) explored the variability of using different ecosystem models with the same or multiple LULC and climate pathways, while Ahlström et al. (2012) and Schaphoff et al. (2006) used climate forcing data from multiple GCMs following the same emission pathway instead of only one climate model to quantify ES indicators. Our sensitivity test using climate inputs from five GCMs and the four LUH1 scenarios indicated uncertainties between 0.1% and 8.6% for the ES indicators considered here (Table S6). The fact that crop type distribution and N fertilization was taken directly from the LUMs for IMAGE and MAgPIE simulations was considered as an extension of the LULC input from the LUMs and therefore as a part of the LULC scenarios of these models. In this study we considered the near to medium future until 2040. Estimates of ES indicator levels and their variability for the far future until 2100 are likely different.

**5.  Conclusions**

We conclude that LUMs and IAMs have some fundamental limitations in capturing all relevant processes related to LULC changes which in some scenarios result in questionable and potentially unrealistic features in their regional LULC allocations and their global and regional trends. More spatial information from LUMs related to regional assumptions (e.g., on technological progress, flows of food

import, export and local production) in addition to LULC patterns would be helpful to better understand their regional LULC allocations. Limitations also include technical aspects of LUMs such as, for instance, adequate spatial resolution, representation of net vs. gross LULC changes, number of LULC classes. We can only deduce a generally high influence of model-specific singularities from our findings, although we couldn't attribute the discrepancy in LULC patterns to the individual factors in the LULC modelling process based on the available LUM data.

The variability across the 16 LULC scenarios entails a high variability in the trends of most ES indicators investigated in this study. On a regional level, this emphasizes the role of tropical forests, and especially the borders of the currently forested tropics, as regions with most uncertain future developments that have the potential to significantly alter regional ecosystem functionality and services in a way that is substantial in a global context. Among the investigated ES indicators, the identification of crop production as the indicator that was associated with the by far highest uncertainties in terms of global totals and regional variabilities, highlights diverging production targets but also different regional strategies of the investigated models to fulfill future demands for crop production.

Our results stress that information from individual LUMs or IAMs, that is used for policy support towards sustainable transformations, should be complemented by further information such as the variability of ES indicators arising through different LULC projections. This would provide a wider context that is essential to be acknowledged in policy making. For regional decision making, this is especially important in regions with highly diverging trends in land-use scenarios and therefore large variability in ES indicators. The issue may become even more relevant in the second half of the century, when LULC changes for climate change mitigation and adaptation are likely to intensify. Ultimately, we need to find improved ways to achieve a better integration between models targeting the different aspects in the cycle of socio-economic developments and their direct, indirect and cumulative implications on natural systems.

**Data availability**

LUH2 data are available at www.luh.umd.edu/data.shtml and ISI-MIP climate data are available at www.isimip.org/gettingstarted/data-access/.

**Author contribution**

A.D. Bayer and A. Arneth conceived and designed the experiments. P.H. Verburg and R. Fuchs made CLUMondo data available and converted them for this study. P. Anthoni prepared LUH2 input data for LPJ-GUESS and assisted in the LPJ-GUESS model simulations. R. Mey, A. Krause and A.D. Bayer carried out the model simulations and led the data analysis with contributions from all authors. R. Fuchs contributed with the analysis of spatial-temporal patterns of land-use models. A.D. Bayer led the writing of the manuscript with contributions from all authors.

**Competing interests**

The authors declare that they have no conflict of interest.

**Acknowledgements**

This work was funded by the European Commission's 7th Framework Program under Grant Agreement numbers 308393 (OPERAs) and 603542 (LUC4C). This work was supported through the Helmholtz Association and its research program ATMO. The authors thank the research groups for making climate model data (ISI-MIP) and land-use model data available to the community. We thank K. Winkler for providing numbers on historical LULC changes.

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

Table 1. Overview of land-use models and scenarios used in this study. Main references are given for each model and the scenarios applied in this study. See also references therein for further details.

| Land-use model | Technical characteristics[1] | LU categories and additional information | Allocation procedure | Consideration of climate change |
|---|---|---|---|---|
| **CLUMondo** (van Asselen and Verburg, 2013; Eitelberg et al., 2016) | S: 9.25 km x 9.25 km, Eckert IV projection. T: future 2001-2040, annual. B: starts from land use in 2000 after (Ramankutty et al., 2008) (Asselen and Verburg, 2012). | 24 land systems types as combinations of extensive/medium intensive/intensive medium intensive cropland, mosaic cropland and grassland, dense/open forest, bare land and built-up area (Note: pigs/poultry are excluded as a defining characteristic). | Spatially explicit allocation of land systems to fulfill the demand of 24 world regions. As long as demands are not satisfied, land systems that contribute more to a standing demand are preferred. No trade between world regions assumed. | None. Present climate and atmospheric $CO_2$ level assumed. |
| **IMAGE** (Popp et al., 2017; Stehfest et al., 2014) | S: 26 world regions (for socio-economic parameters). 0.5°x0.5° (most environmental parameters, incl. land use) T: historic 1970-2005, future 2005-2100, annual. BL: LU harmonized to HYDE 3.1 (Klein Goldewijk et al., 2011) in 2005[1]. | Fractional data for cropland, pasture, forest, urban, other natural. | Allocation relies on regression-based suitability assessment and iterative allocation procedure until demands are met. | Yes. All scenarios base on RCP2.6 climate from IPSLCM5A-LR general circulation model bias corrected to the 1960-1999 historical period. Varying $CO_2$ level with 490 ppm in 2100. |
| **MagPIE** (Lotze-Campen et al., 2008; Popp et al., 2014, 2017) | S: 10 world regions, (0.5° x 0.5° resolution, clustered to 500 units based on 'similarity' in the modeling, see Humpenöder et al., 2014). T: future 1995-2100, 5 year steps, B: LU harmonized to HYDE 3.1 (Klein Goldewijk et al., 2011) in 1995. | Fractional data for cropland (irrigated, non-irrigated), pasture, forest, urban, other natural. | Achieve demands under minimizing costs in 10 world regions with recursive dynamic optimization. Cost minimization defines land availability. International trade considered (Humpenöder et al., 2014). | Yes. All scenarios base on RCP2.6 climate impacts same as IMAGE. |
| **LUH1** (Hurtt et al., 2011) | S: 0.5°x0.5°, WGS84, T: historic 1500-2005, future 2005-2100, annual. B: historic product is based on (Klein Goldewijk et al., 2011). | Fractional data for cropland, pasture, primary vegetation, secondary vegetation, urban. Gross transitions between land-use classes based on land-use change: -Shifting cultivation cultivation in some tropical regions. | Allocation for historic period based on HYDE 3.1 (land-use/capita using weighing maps for built-up area, population density, soil suitability, coastal areas/river plains, annual mean temperature). Additionally, LUH1 uses the following features to process land use change: -Shifting cultivation rates -Priority for land conversion (agricultural land included in wood harvest statistics -Deforestation for agricultural land taken from primary or secondary) For the future, the allocation is dependent on the four IAMs used for the scenarios (at 2° x 2° resolution), disaggregated to 0.5° x 0.5°. | Yes. Consideration of different Representative Concentration Pathways (RCP's) from the different IAMs. |
| **LUH2** (Hurtt et al., 2020, data from http://luh.umd .edu) | S: 0.25°x0.25°, WGS84, T: historic 850-2015, future 2016-2100, annual. B: 850–2015 land-use based on HYDE 3.2 (Goldewijk et al., 2017). | Fractions for cropland (C3/C4 annuals, C3/C4 perennial, C3 nitrogen fixing), managed pasture, rangeland, primary land, secondary land, urban. | Allocation for historic period based on HYDE 3.2 and globally available statistics and for the future period based on transitions and additional information from IAMs (see also their respective allocation procedures, compare references in (Hurtt et al., 2020). Additionally, LUH2 uses the following features to process land use change: - crop type and rotations - Shifting cultivation rates - Historical wood harvest statistics and future forest transitions - Biomass density and recovery rates - Priority for land conversion. | Yes. Consideration of different RCPs from the different IAM's. |

| Scenarios |
| --- |
| • FAOref: demands of tons of crop production, size of livestock and population (as built-up area) as expected by FAO until 2040. Demands projected to 24 world regions by the integrated assessment model IMAGE.<br>• Cstor: in addition to FAOref extra demand for C storage implemented with a no-carbon-loss approach. This results in the favorable consideration of land systems with high C storage capacity such as dense forests in the allocation procedure.<br>• Bdiv: in addition to FAOref extra demand for protected areas in the allocation procedure. Protected areas used that follow national biodiversity targets according to Aichi target number 11. Forest and grassland proportions of natural, semi-natural and managed natural land-systems were selected to account for protected areas in CLUMondo land systems in allocation procedure. Land systems in land areas assigned to IUCN categories I to IV at the beginning of the simulation procedure were maintained throughout the scenario. Favorable consideration of land systems with high amount of protected areas dense forest, mosaic grassland and forest, mosaic grassland and bare, natural grassland. |
| • Base: land-use change driven by increased food demand and population growth according to SSP2.<br>• ADAFF: Demands as in Base scenario with additional CDR target of 130 GtC by 2100. Land-based mitigation achieved by avoided deforestation and afforestation.<br>• BECCS: Demands as in Base scenario with additional CDR target of 130 GtC by 2100. Land-based mitigation achieved by bioenergy plant cultivation and subsequent carbon capture and storage. |
| • Base: land-use change driven by increased food demand and population growth according to SSP2.<br>• ADAFF: Demands as in Base scenario with additional CDR target of 130 GtC by 2100. Land-based mitigation achieved by avoided deforestation.<br>• BECCS: Demands as in Base scenario with additional CDR target of 130 GtC by 2100. Land-based mitigation achieved by bioenergy plant cultivation and subsequent carbon capture and storage. |
| • 26Be: Scenario following RCP 2.6 where the cultivation of bioenergy crops with CCS sequestering carbon that significantly balances C so that global warming saturates below 2°C until 2100. Increased land demand for bioenergy crops, which occur near existing agricultural areas, natural areas are declining. Simulated by IMAGE IAM.<br>• 45Aff: Scenario following RCP 4.5 assuming that global GHG emission prices support climate change mitigation by massive afforestation programs around the world with the aim to preserve large C stocks in forests. Vast expansion of forested area slightly decreases the area of arable land, but assumed technological improvements and more efficient global trading systems keep up agricultural production to satisfy demands. Simulated by dynamic recursive economic model GCAM.<br>• 60Stab: Scenario following RCP 6.0 assuming stabilization with a medium population growth throughout the 21st century and climate change mitigation efforts only become effective in the last third of the century. Intensification and expansion of cropland area cover the increased food demand. Simulated by AIM and other models.<br>• 85Pop: Scenario following RCP 8.5 assuming rapid population growth to 12 billion people in 2100 and a global temperature rise by more than 4 °C until 2100. Intensification is supported by technological improvements as the most important driving factor to satisfy a high food demand. Cropland expands as well as pasture especially in developing countries, while natural areas decline. Simulated by MESSAGE IAM. |
| • SSP1-26: Scenario combining SSP 1 and RCP 2.6 of an environmentally friendly world with low population growth, high urbanization, relatively low demand for animal products and high agricultural productivity. Land use and land use change is relatively low and policies targeting the reduction of atmospheric greenhouse gases (including BECCS and afforestation) are implemented so that an additional forcing is limited to 2.6 W m$^{-2}$ by 2100. Simulated by IMAGE 3.0 IAM where LULC is allocated iteratively based on suitability assessment until demands are met.<br>• SSP3-70: Scenario combining SSP 3 and RCP 7.0 focused on regional development. High population growth in developing countries, slow economic development and per capita food consumption lead to high expansion of agricultural areas rather than agricultural land intensification under relatively high level of climate change with a radiative forcing of 7.0 W m$^{-2}$ by 2100. Simulated by AIM/GCE IAM framework working based on the adjustment of prices until supply and demand for commodities and services equilibrate.<br>• SSP5-85: Scenario combining SSP 5 and RCP 8.5 of a world characterized by strong economic growth that is based on fossil fuels, with low population growth, high urbanization, and high food demand per capita with high agricultural productivity due to technological progress. Strong expansion of global cropland and the level of climate change is high with an assumed radiative forcing of 8.5 W m$^{-2}$ by 2100. Simulated by REMIND/MAgPIE IAM framework where LULC changes are modelled chiefly based on prices and quantities of bioenergy and greenhouse gas emissions. |

¹S: spatial resolution and projection, T: time horizons, B: underlying data bases or historical land-use baseline from which scenarios start, ²small
deviations in the area of the land-cover classes between MAgPIE/IMAGE and HYDE 3.1 land-use in 1995/2005 occurring due to different land
masks and calibration routines.

Table 2. Total area of cropland, pasture and natural land and change therein from 2000-2004 to 2036-2040 for 16 land-
use scenarios. For each scenario, the left column gives the global total area for 2000-2004 (upper) and 2036-2040
(lower) and the right column gives the change from 2000-2004 until 2036-2040 in absolute terms (upper) and in %
relative to the level in 2000-2004 (lower). Grey shading indicates a positive or negative trend. Total area under change
from 2000 to 2040 is given in absolute terms and as % of the global ice-free land area considered in this study (see
methods). Minor deviations in numbers may occur due to rounding.

| | Cropland [$10^6$ km$^2$] | | Pasture [$10^6$ km$^2$] | | Natural [$10^6$ km$^2$] | | Total area under change between 2000 and 2040 [$10^6$ km$^2$] |
|---|---|---|---|---|---|---|---|
| **CLUMondo FAOref** | 15.4 17.5 | +2.1 +13.7% | 27.4 26.5 | -0.9 -3.3% | 89.3 88.1 | -1.2 -1.3% | 9.4 7.1% |
| **CLUMondo Cstor** | 15.4 17.3 | +1.9 +12.3% | 27.4 26.2 | -1.2 -4.3% | 89.4 88.7 | -0.7 -0.8% | 10.2 7.7% |
| **CLUMondo Bdiv** | 15.4 17.1 | +1.7 +11.3% | 27.4 26.4 | -1.0 -3.8% | 89.4 88.7 | -0.7 -0.8% | 12.7 9.6% |
| **IMAGE Base** | 15.6 16.2 | +0.6 +3.8% | 35.8 38.6 | +2.8 +7.7% | 80.8 77.4 | -3.4 -4.2% | 10.5 8.0% |
| **IMAGE ADAFF** | 15.6 15.9 | +0.3 +1.9% | 35.8 35.0 | -0.8 -2.3% | 80.8 81.3 | +0.5 +0.7% | 9.8 7.4% |
| **IMAGE BECCS** | 15.6 18.1 | +2.5 +15.9% | 35.8 38.6 | +2.8 +7.8% | 80.7 75.4 | -5.3 -6.5% | 12.8 9.7% |
| **MAgPIE Base** | 15.8 19.2 | +3.4 +21.6% | 33.4 31.6 | -1.7 -5.2% | 83.0 81.3 | -1.7 -2.0% | 5.1 3.9% |
| **MAgPIE ADAFF** | 15.8 18.6 | +2.7 +17.4% | 33.4 31.0 | -2.4 -7.2% | 83.0 82.6 | -0.3 -0.4% | 6.0 4.5% |
| **MAgPIE BECCS** | 15.8 20.0 | +4.2 +26.4% | 33.4 31.3 | -2.4 -7.2% | 83.0 80.9 | -2.1 -2.6% | 6.0 4.6% |
| **LUH1 26Be** | 15.2 19.1 | +3.9 +26.1% | 33.2 32.4 | -0.8 -2.5% | 83.8 80.7 | -3.1 -3.7% | 8.5 6.4% |
| **LUH1 45Aff** | 15.2 12.9 | -2.3 -15% | 33.2 29.0 | -4.2 -12.5% | 83.8 90.3 | +6.4 +7.7% | 8.9 6.7% |
| **LUH1 60Stab** | 15.2 16.0 | +0.9 +5.6% | 33.2 25.7 | -7.5 -22.7% | 83.8 90.5 | +6.7 +8.0% | 15.1 11.4% |
| **LUH1 85Pop** | 15.2 16.6 | +1.4 +9.6% | 33.2 35.2 | +2.0 +6.1% | 83.8 80.4 | -3.5 -4.1% | 5.9 4.5% |
| **LUH2 SSP1-26** | 14.9 15.3 | +0.3 +2.1% | 32.8 30.8 | -2.1 -6.2% | 79.8 81.5 | +1.7 +2.2% | 10.6 8.0% |
| **LUH2 SSP3-70** | 14.9 17.4 | +2.5 +15.5% | 32.8 33.8 | +1.0 +2.9% | 79.8 76.4 | -3.4 -4.3% | 13.7 10.3% |
| **LUH2 SSP5-85** | 14.9 17.7 | +2.7 +18.3% | 32.8 31.6 | -1.2 -3.7% | 79.8 78.3 | -1.5 -1.9% | 10.4 7.8% |
| **Average and uncertainty across 16 LU scenarios** | 15.4 ± 0.3 17.2 ± 1.8 +1.8 ± 1.6 +11.7% ± 10.5% | | 32.6 ± 2.8 31.5 ± 4.1 -1.0 ± 2.6 -3.5% ± 7.6 % | | 83.4 ± 3.4 82.7 ± 5.0 -0.7 ± 3.3 -0.9% ± 4.0% | | 9.7 ± 2.9 7.4% ± 2.2% |

257

258

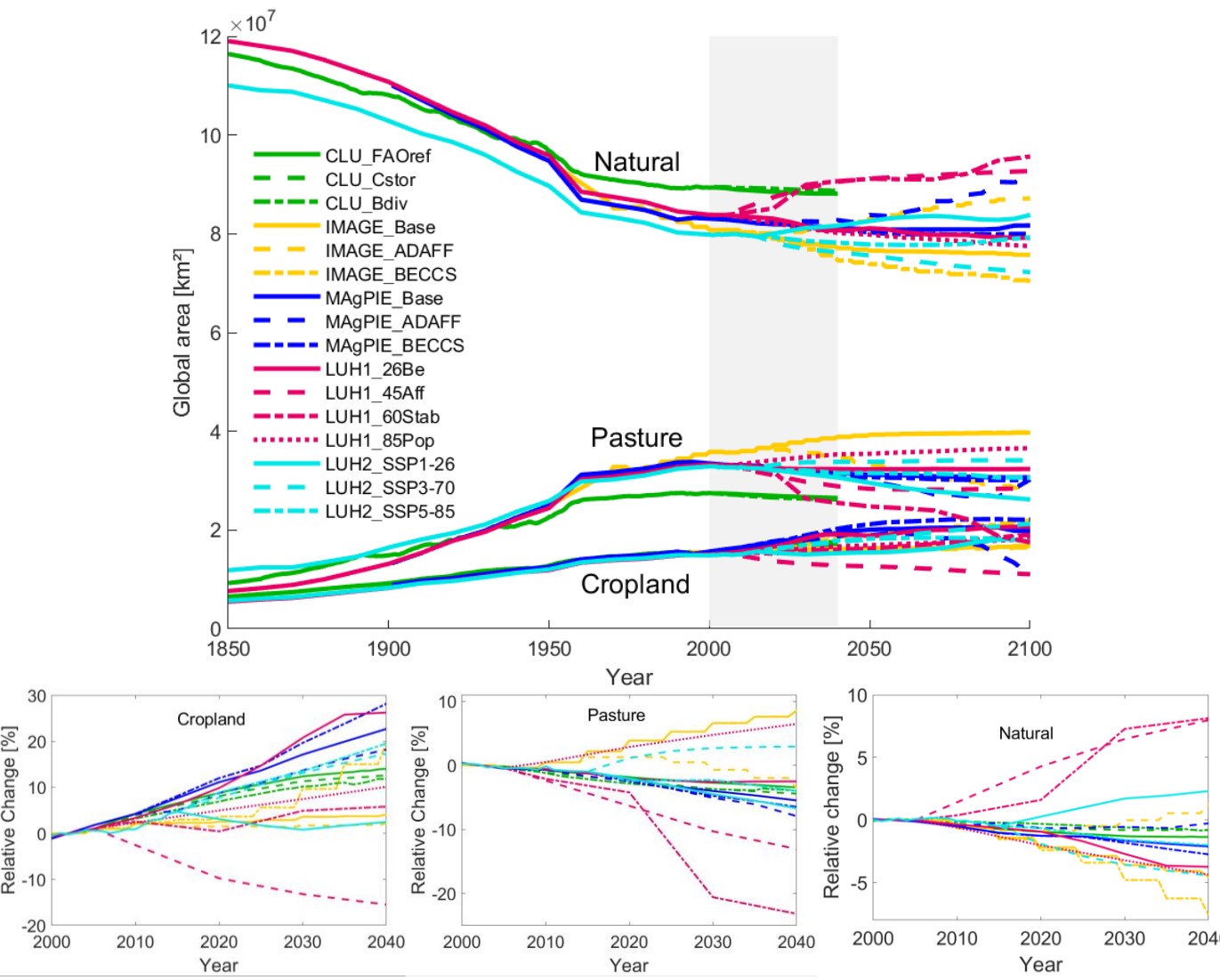

259

260

Fig. 1. Absolute land area of croplands, pastures and natural areas between 1850 and 2100 for 16 scenarios of five land-use models and detailed relative changes in LU from 2000 to 2040 analyzed in this study.

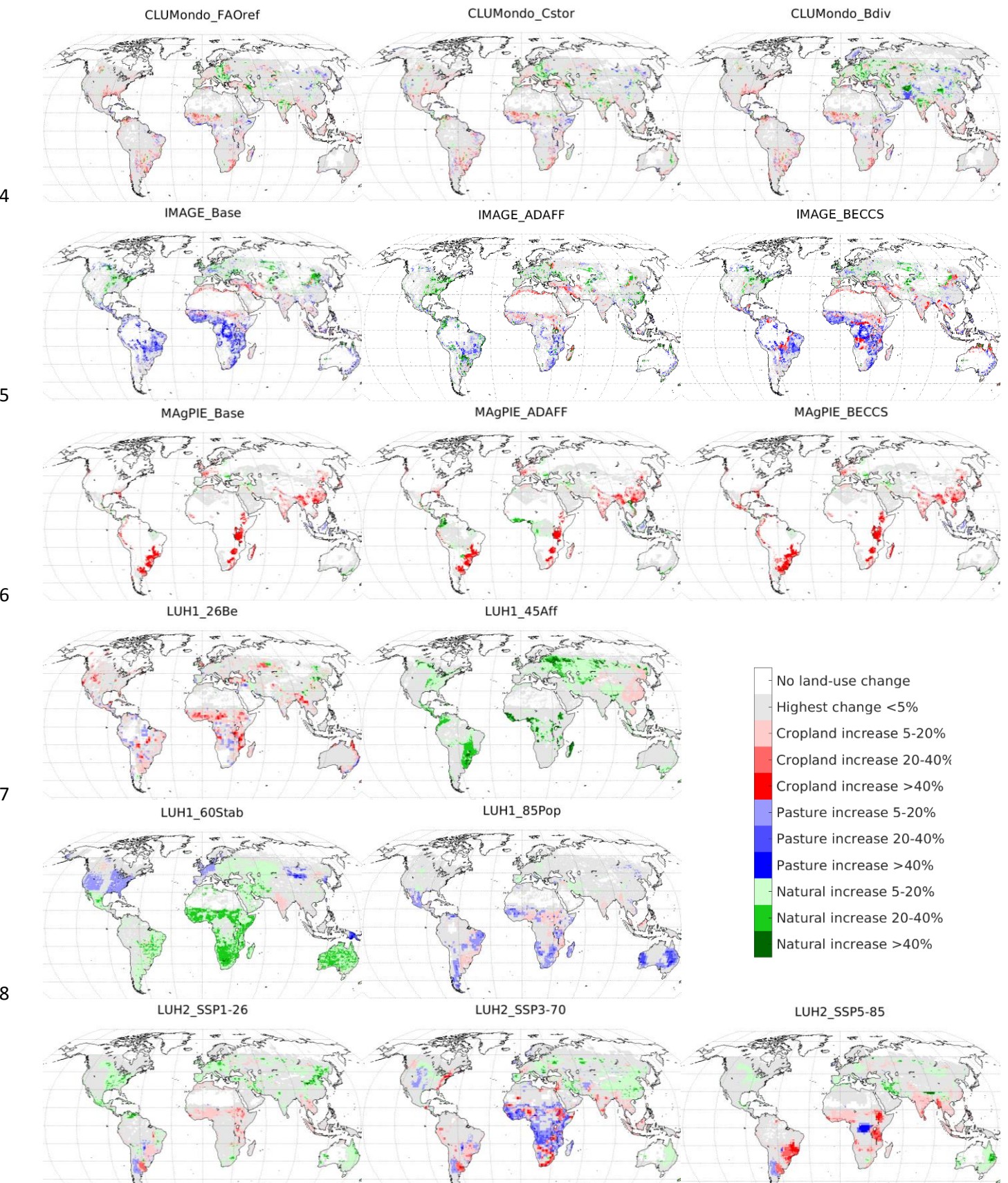

Fig. 2. Categories of dominant land-use change from 2000-2004 to 2036-2040 for each of 16 land-use scenarios. The legend is identical for all plots.

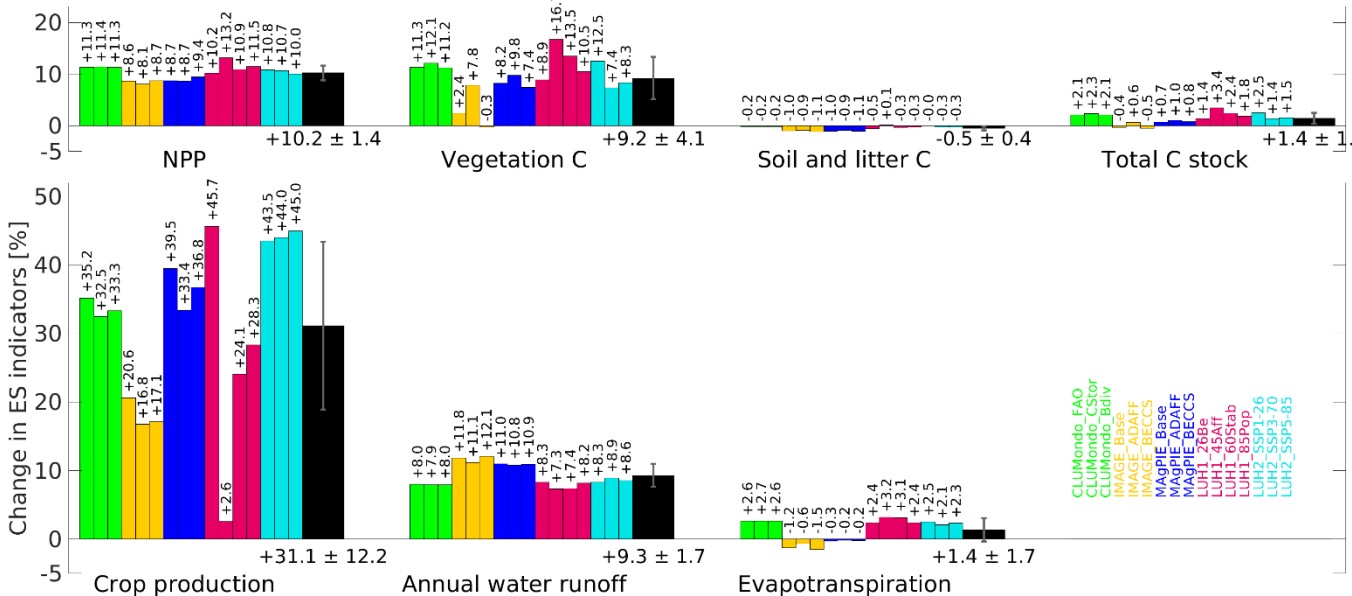

272

Fig. 3. Change and uncertainty in ES indicators from 2000-2004 to 2036-2040 in percent relative to the base level in
2000-2004 across 16 land-use scenarios. Black bars give average and standard deviation of the relative changes across
all scenarios. See Table S3 for absolute levels and changes for each scenario.

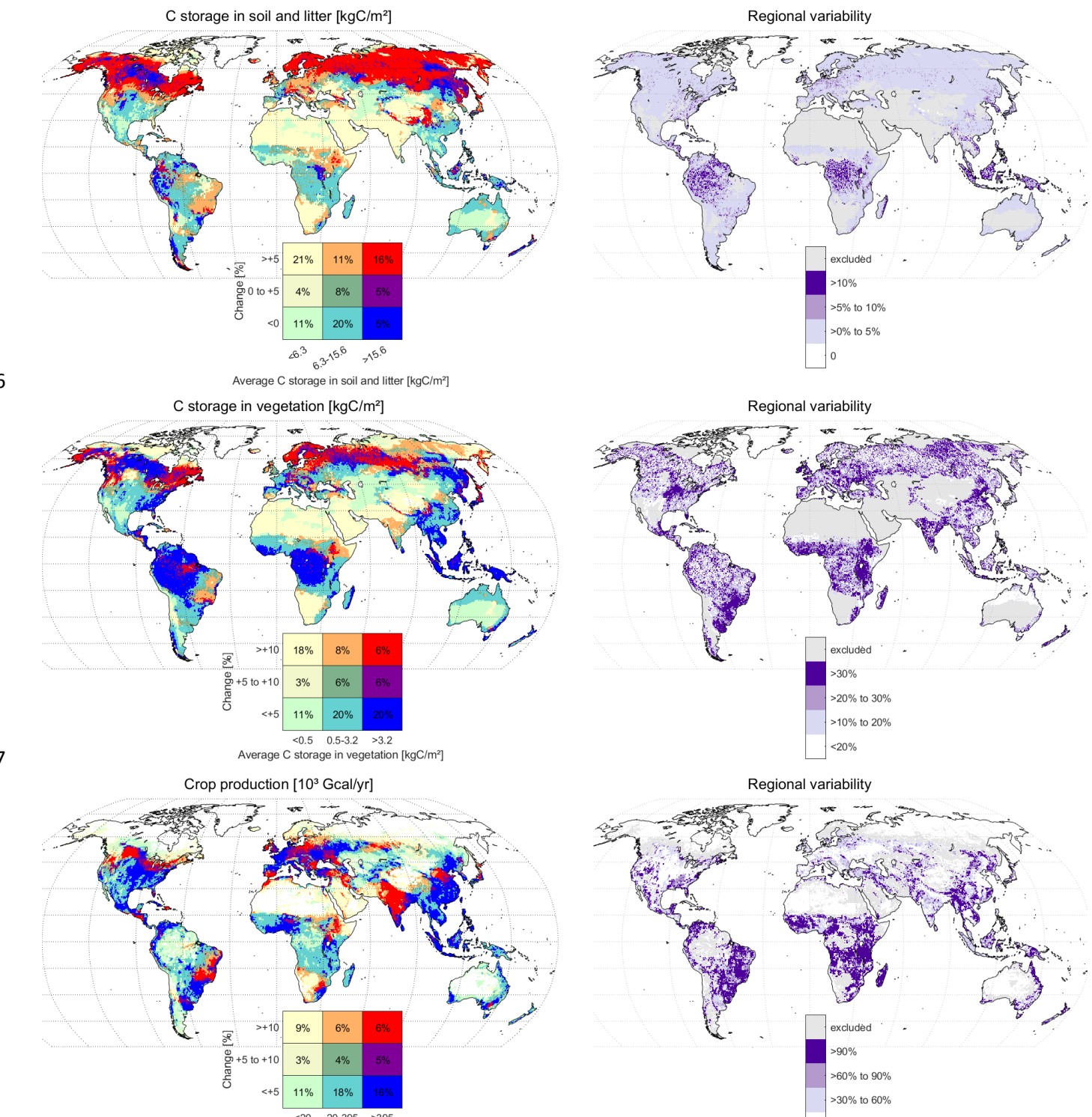

276

277

278

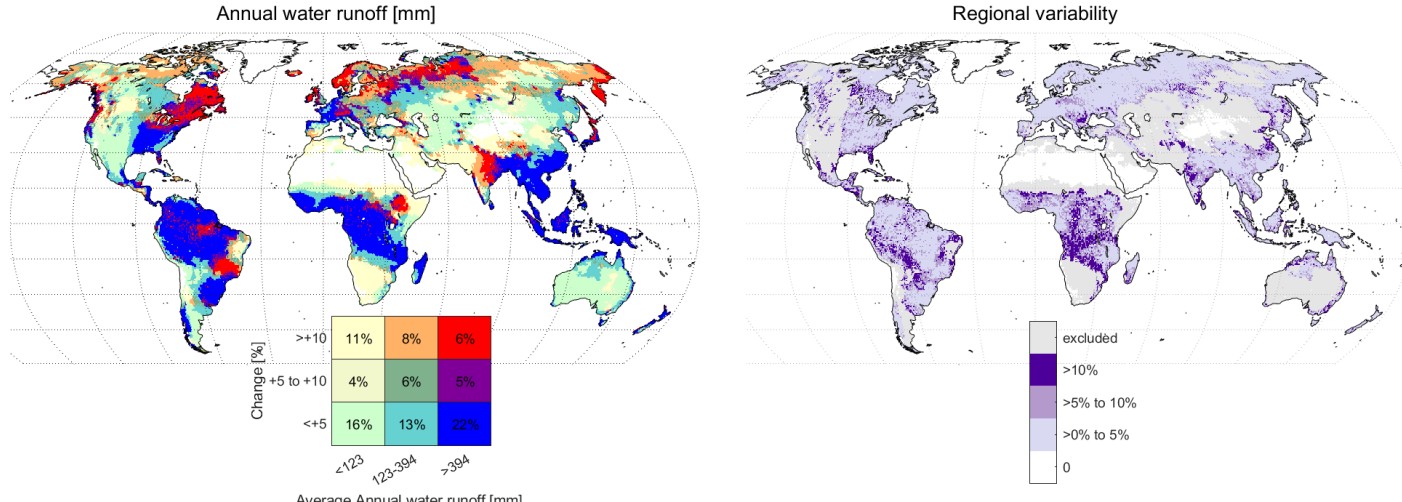

Fig. 4. Left maps show categories of the average level of the provision of selected ES indicators in 2000-2004 and the relative change until 2036-2040 averaged over 16 land use scenarios. Thresholds for categories for the average ES indicator level follow 33[rd] and 67[th] percentiles for each ES indicator while the change in indicators is given is 5% steps for all indicators to allow for comparability. Note that for soil and litter C the lowest category is negative. At high levels of ES indicator provision (highest 33% of values), blue cells mark regions where on average little change and red cells where high changes are expected until 2040. The yellow category marks regions where base levels in ES provision are low. Therefore, relative changes in them can be very high but are of minor importance. The percentage of global land area in each category is indicated in %. Cells where the average indicator level in 2000-2004 is 0 are colored white and excluded from the statistics. Right maps give the variability of the percent change in each ES indicator for each cell which was calculated as the standard deviation of the changes in the ES indicator from 2000-2004 to 2036-2040 that was derived for each of the 16 land use scenarios individually. Regions, where the base level in 2000-2004 was below the 33[rd] percentile (yellow to green cells in left maps) were excluded in regional variability maps and colored grey to focus on cells with relevant ES indicator provision. Note that the legend scaling is different for vegetation C and crop yield production. Purple regions indicate a standard deviation in the predicted relative changes of this indicator higher than 10% of the indicator level in 2000-2004 (30% for vegetation C, respectively, 90% for crop yield production). See SI Fig. S2 for NPP, total C storage and evapotranspiration.