# Peer review of "Diverging land-use projections cause large variability in their impacts on ecosystems and related indicators for ecosystem services"

_Earth System Dynamics, 2020_

## Short Comment (SC1) · 1 Jul 2020

This study analyzes 16 scenarios from five land use models (LUMs) and integrated assessment models (IAMs) and the effects of the resulting land use changes towards 2040 on ecosystem service (ES) indicator values. The text is easy to read, the work in well-embedded in existing literature, and the results are visualizaed comprehensively. I have two main questions/concerns with respect to this study, as detailed below.

Model-coupling mismatches: To my understanding, you use a one-way coupling between the LUM/IAM and LPJ-GUESS. This can lead to mismatches. For example, the IAM has computed production of particular agricultural products for a region, and con-

[Figure]

verted this into a cropland area per grid cell. In LPJ-GUESS, this grid cell contains a prescribed fraction of crop functional types, which can mismatch with the products in the IAM, and a certain yield, which can mismatch with the production (supply) of agricultural area of that grid cell in the IAM. I believe that such mismatches have a large effect on the results. I expect that large differences in ES indicator values between two models will occur when one of them has a good match with LPJ-GUESS (due to the use of the same data sources) and another a bad match. The comparison between ES indicator values is not fair in this case, because for the second model, its assumptions are violated by the LPJ-GUESS model (for example, demand in the IAM does not match supply computed by LPJ-GUESS). I think it would help to explain how the models are coupled exactly (what variable(s) is/are exchanged) and to bring this point up in your discussion if you agree with this potential issue.

Scenario projections: It is recognized that LUMs/IAMs do not provide predictions (as weather predictions do), but instead projections, meaning that they are conditional (what if ... ?). What conditions are evaluated depends on the question at hand. The conditions can range from realistic (business as usual) to very irrealistic. Irrealistic scenarios can still be useful as thought experiments, to better understand the system, to serve as warnings for worst-case effects, or to evaluation potential policy interventions. As such, in lines 487 and further, you discuss that some scenario results don't seem plausible. But perhaps they aren't meant to be plausible. Therefore, given that the conditions, and thus the scenarios, depend very much on the question asked in the original study, it is not clear to me what you are exactly evaluating when assessing the variation is ES indicator outcomes over all these models and scenarios combined. I could see the added value of analyzing ES indicator outcomes of all business as usual scenarios, as that would show the effect of different assumptions about the working of the current system on ES impacts, but the value of comparing among the other scenarios (which could have easily been very different if a different questions were asked) is not quite clear to me. In the current version of the manuscript, you only comment about this seems to be "However, conclusions drawn here in regard to projected changes

in LULC and ES indicators are inherently dependent on the selected set of LUMs and scenarios, evaluation time period and simulation set-up", which does not really help the reader to see what can and cannot be learned from the results given this dependence.

Minor comments:

481-486: In the context of this paragraph, which speculates about the potentially more valid small-scale changes of some of the models, you may be interested to know that, in a recent study of LUC in Brazil, we found that indeed the small-scale changes of a spatially-explicit LUM were more accurate than the larger-scale changes of an economic model, see Stepanov et al. 2020, doi:10.3390/land9020052

691-692: "We conclude that LUMs and IAMs have fundamental limitations in capturing all relevant processes related to LULC changes." I don't see how your results lead to this conclusion.

---

## Referee Comment (RC1) · Anonymous Referee #1 · 18 Aug 2020

This study analyzes an ensemble of future land use projections arising from a set of 5 LUMs/IAMs. These land use projections are then translated into changes in ecosystem services (ES) using the LPJ-GUESS model. One of the main conclusions is that there is a large spread in land use projections (and therefore ES) and that most of this spread originates from structural model differences (i.e. choice of LUM/IAM) rather than from socio-economic assumptions (i.e. choice of scenario storyline within a given model). Overall, this is an interesting study which makes an important point about the large uncertainties associated with future land use projections and their potential causes. However some aspects of the manuscript could still be improved and require additional clarifications.

[Figure]

I found the whole discussion about the role of the baseline level very hard to follow. I understand from the analysis that there are very different starting states in 2000-2004 across the models. But does it affect the conclusions of the study or not? One could for instance intuitively expect that in a model starting with lower present-day natural vegetation there would more room for future expansion. More specifically, in which way is the baseline level taken into account when looking at relative changes (L236-239)? And where is the assumption that "Effects of differences in the modeling protocol of CLUMondo/LUH1/LUH2 and IMAGE/MAgPIE simulations for projections of ecosystem dynamics affect especially the base level of ES indicators in 2000-2004 and the relative deviations over time, which this study is focused on, only to a small degree" coming from? This last sentence by the way is so convoluted that I might just have misinterpreted its meaning. On that same issue, it is somewhat disturbing that the IPSL-based climate forcing has been bias-corrected, implying a sort of harmonization between observed and projected climate, while no harmonization was performed for the land use forcing. Why not following the same logic for both climate and land use?

I have been wondering whether the positive trend in some ES indicators could be affected or even reversed if fires were properly accounted for. Referring to fire and other processes, section 4.2 has a rather elusive statement: "Because these processes are only to some degree implemented in LPJ-GUESS (see, e.g., Pugh et al., 2019), this could further increase the regional variability in ES indicators as indicated in this study." Something more explicit would be welcome, such as clarifying upfront in section 2.2 how fires are accounted for in the model.

The conclusion that "some scenarios show questionable and possibly unrealistic features in their LULC allocations" could warrant some more in-depth evaluation of historical trends to be fully supported. Although this might be an ambitious task, the discussion could at least outline some evaluation strategies that should be deployed in future studies in order to pinpoint more specific deficiencies.

I am still confused about the procedure by which crops are prescribed in LPJ. The

total crop fraction evolves according to the given land use scenario, I assume, while the particular mixture of crop types is prescribed to be constant in time after 2006 (table S1). If this mixture is constant in time, then how was it possible to represent crop adaptation by "simulating the adequate selection of suitable crop varieties under changing climate" (section 2.2)?

Could you please clarify if NPP is aggregated over all ecosystem types including crops? I suppose this is the case, which would imply that there is some information overlap between the NPP and crop productivity indicators. Would it be possible to show some disaggregated results for NPP (i.e. separately for the 3 types of vegetation represented in LPJ)? It might help to reveal some ecosystem-specific responses.

L31-33: please provide an uncertainty range for all variables along with the median value.

L229: Could you clarify how C storage from CCS was quantified, given that BECCS is not represented in LPJ (table S1)?

L244: it would be nice to include table S2 in the main text.

L256-260: this part would fit better in the method section. Moreover, it would be good to include a cross-walking table in the method section to explain how the land classes of the respective LUMs/IAMs were translated into the 3 main types in LPJ. i.e., this needs to be clarified not only for CLUMondo.

Fig 3: please add the units for all variables.

---

## Referee Comment (RC2) · Anonymous Referee #2 · 20 Aug 2020

General comments

Bayer et al. presented a comparison of multiple land use and land cover (LULC) scenarios broadly used for understanding the future impacts of global change on carbon emission and ecosystem functioning. They highlight the large discrepancies amongst scenarios and what are the implications of this broad variety of scenarios on the trends of several ecosystem service variables. LULC is unquestionably an important driver of future changes and understanding and assessing the uncertainty and variability of such scenarios is critical for both science and policy makers. Therefore, the topic of this study is relevant and timely, and this work can be an important contribution, but

it still requires some clarifications in the methodological approach and the interpretation of results, and potentially one additional analysis that should be straightforward to implement and could make the manuscript more appealing.

My main concern with the current presentation is that the authors highlight the large discrepancies amongst LULC scenarios and even mention that some scenarios predict unrealistic regional patterns of LULC change. However, I understood that the authors used only one scenario (RCP2.6) from one model (IPSL-CM5A-LR), whereas many of the LCLU scenarios were aligned to a broad range of future scenarios, and potentially had very different initial states that may explain at least some of the discrepancies. Couldn't these differences explain some or most of the discrepancies amongst scenarios? Also, the discrepancy across LULC is not necessarily a bad thing, as many of the scenarios selected by the authors are for different socio-economic pathways and they should be different. The authors could clarify these points in a revised version.

Also, the authors presented and discussed the variability in the trajectory of ecosystem services (ES) across all scenarios. This is fine but I think an analysis comparing the emergent responses across all the scenarios analyzed could give much more insight on how LCLU change could affect the future of ecosystem functioning. For example, the authors could relate the changes in cropland/forest/pasture area in each continent with changes in runoff or evapotranspiration and use the slope of these relationships to understand the sensitivity of ecosystem services to LCLU changes.

Specific/Minor comments

Line 29. I don't think abrupt transitions necessarily indicate problems in this case. For example, a tropical forest may be minimally disturbed until new infrastructure (e.g., paved roads) is built. Likewise, policies change with governments and can result in significant reversals of trends, for example, the significant reduction in deforestation in Brazil in the 2000s and the current increase.

Line 35. I suggest to replace "edge" with boundaries or ecotones. Forest edge is

commonly use in landscape scale to define transitions from deforested and forest areas at landscape level (sensu Skole and Tucker 1993). Same comment for Line 75.

Line 80. Rephrase: large uncertainties in LCLU projections may affect the confidence in projected changes in ecosystem functioning, but not the ecosystem functioning per se.

Line 115. Some word is missing after "following".

Section 2.2. Maybe I missed it, but what happens when the LCLU scenarios are inconsistent with the LPJ vegetation? For example, if the scenario indicates timber harvesting but LPJ does not predict any forest? In fact, it is unclear to be whether or not logging was considered in these simulations.

Line 195. What was the rationale for selecting RCP2.6 instead of other pathways? It seems to me that RCP2.6 is rather too optimistic.

Line 209. Related to my point in section 2.2. It is fine to group all the natural classes, but this still does not clarify what happens in the case of logging.

Line 231. I understand the rationale for minimizing the role of interannual variability, but it is unclear to me that 5 years is sufficient. Would the results change considerably if, for example, 10-year averages were used instead?

Section 3. The authors have a separate discussion section, but the text in the results section often read more like discussion (e.g., most of the paragraph starting in line 391). Also, I think a multi-panel figure that showed the average changes in cropland, pasture, and forest (both increase and decrease) would help to summarize the results.

Line 273. This sentence is confusing.

Line 288. Examples of some countries?

Line 372. "Central" instead of "Middle"?

Lines 391–422. Can changes in irrigation also contribute to changes in ET and runoff in LPJ-GUESS? Does LPJ-GUESS simulate irrigation?

Line 464. It was more than soy moratorium in the case of Brazil, law enforcement and policy changes were also important (Nepstad et al. 2014).

Line 638. I suggest "South American" Cerrado and Chaco instead to remove ambiguity, as the Chaco is not in Brazil.

Line 709. I agree with this paragraph and this is why I also suggested the analysis on the emergent responses. It seems that the authors already have the results ready for at least some initial analysis to qualify the changes in ES responses as functions of LULC changes.

Figures 1 and 4. Some of the colours are difficult to distinguish, at least for me (CLU and LUH2, for example in Fig. 1). For Figure 4, the authors could fix hues for different levels of relative change (rows), and fix brightness for the absolute values (columns), it would also make the figure more intuitive.

References

Nepstad, D., D. McGrath, C. Stickler, A. Alencar, A. Azevedo, B. Swette, T. Bezerra, M. DiGiano, J. Shimada, R. Seroa da Motta, E. Armijo, L. Castello, P. Brando, M. C. Hansen, M. McGrath-Horn, O. Carvalho, and L. Hess, 2014: Slowing Amazon deforestation through public policy and interventions in beef and soy supply chains. Science, 344 (6188), 1118–1123, doi:10.1126/science.1248525.

Skole, D., and C. Tucker, 1993: Tropical deforestation and habitat fragmentation in the Amazon: Satellite data from 1978 to 1988. Science, 260 (5116), 1905–1910, doi:10.1126/science.260.5116.1905.

---

## Author Comment (AC1) · 2 Dec 2020

**Reply to anonymous referee #1**

*This study analyzes an ensemble of future land use projections arising from a set of 5 LUMs/IAMs. These land use projections are then translated into changes in ecosystem services (ES) using the LPJ-GUESS model. One of the main conclusions is that there is a large spread in land use projections (and therefore ES) and that most of this spread originates from structural model differences (i.e. choice of LUM/IAM) rather than from socio-economic assumptions (i.e. choice of scenario storyline within a given model). Overall, this is an interesting study which makes an important point about the large uncertainties associated with future land use projections and their potential causes. However some aspects of the manuscript could still be improved and require additional clarifications.*

We thank the reviewer for the expressed interest in our manuscript. In the revisions to the manuscript we will be addressing the raised questions as described below.

*I found the whole discussion about the role of the baseline level very hard to follow. I understand from the analysis that there are very different starting states in 2000- 2004 across the models. But does it affect the conclusions of the study or not? One could for instance intuitively expect that in a model starting with lower present-day natural vegetation there would more room for future expansion. More specifically, in which way is the baseline level taken into account when looking at relative changes (L236-239)? And where is the assumption that "Effects of differences in the modeling protocol of CLUMondo/LUH1/LUH2 and IMAGE/MAgPIE simulations for projections of ecosystem dynamics affect especially the base level of ES indicators in 2000-2004 and the relative deviations over time, which this study is focused on, only to a small degree" coming from? This last sentence by the way is so convoluted that I might just have misinterpreted its meaning. On that same issue, it is somewhat disturbing that the IPSL-based climate forcing has been bias-corrected, implying a sort of harmonization between observed and projected climate, while no harmonization was performed for the land use forcing. Why not following the same logic for both climate and land use?*

Starting with the last question first: the main reason for using a bias corrected climate is not so much the seamless historical-future transition (which is essential for e.g. carbon and water cycle simulations: without a harmonized climate time series, an artificial offset would be created between the end of the historical and the beginning of the projected future climate), but the fact that simulated climate output from GCMs is much more reliable in terms of anomalies, rather than in terms of absolute values – most GCMs are biased compared to "real" climate values in a grid cell. Therefore, projected raw daily GCM output has to be corrected, using the differences in the mean and variability between GCM and observations in a reference period.

The reviewer is correct that in principle one could also do that for land-use historical-to-future timeseries. However, doing so would to large degree affect the underlying objectives of our study. We aim to show here the large uncertainty/variability in terms of impacts on ecosystems that arises both from the very different (unknown) socio-economic futures as well as from the different LUC modelling approaches. This includes keeping the differences in the historical baseline, although the latter is less prominent in the manuscript by us concentrating on the relative changes. We will clarify this in the manuscript upon revision, and by doing so will also revise the sentence "Effects of…", which is indeed poorly phrased.

Suggested revisions: „The differences in the modeling protocol of CLUMondo/LUH1/LUH2 and IMAGE/MAgPIE simulations will affect the base level of ES indicators in 2000-2004 to some degree, although the impacts of slightly different historical model periods and spin-up and historical climate would have diminished by the beginning of the 21st century (baseline period). Larger effects arise from the differences in the individual land-use change models per se (see also Alexander et al., 2017). In principle, differences in the baseline land-cover maps could spill-over to the degree of change in the future scenarios. For instance, presence or absence of natural vegetation in the base-line maps might

translate into different degrees of future (semi)natural vegetation re-growth. However, this would only be an important consideration for similar scenarios (and their underpinning storylines related to e.g. sustainability). By contrast, the alternative approach of harmonizing the different projections to the same starting point of land-cover would artificially mask some of the simulated differences in ecosystem services indicators which would be contrary to our objectives."

*I have been wondering whether the positive trend in some ES indicators could be affected or even reversed if fires were properly accounted for. Referring to fire and other processes, section 4.2 has a rather elusive statement: "Because these processes are only to some degree implemented in LPJ-GUESS (see, e.g., Pugh et al., 2019), this could further increase the regional variability in ES indicators as indicated in this study." Something more explicit would be welcome, such as clarifying upfront in section 2.2 how fires are accounted for in the model.*

LPJ-GUESS, like other DGVMs, models fire explicitly, but has a simplified representation of other forms of disturbances (which can arise from storms, or insect attaches, etc.). We will clarify this both in the methods, as well as in section 4.2.:

"Because only fire is explicitly simulated as a stand-replacing disturbance process in LPJ-GUESS, while others are subsumed in a stochastic background-disturbance, this could further increase the regional variability in ES indicators (see, e.g., Pugh et al., 2019)."

*The conclusion that "some scenarios show questionable and possibly unrealistic features in their LULC allocations" could warrant some more in-depth evaluation of historical trends to be fully supported. Although this might be an ambitious task, the discussion could at least outline some evaluation strategies that should be deployed in future studies in order to pinpoint more specific deficiencies.*

The reviewer is certainly right that a full evaluation of pros and cons of the implementation of scenarios' storylines is beyond our study. We aspired to put scenarios' projected LULC changes in a historic context because this is our most realistic reference which led us us to the presented conclusion. We suggest to add the following text to the manuscript in order to take up the suggestions (section 4.1.);

"In this context it seems worthwhile for the land use modelling community to evaluate future land use changes against historic trends, in spatial, temporal and thematic aspects. This may help to avoid some of the deemed questionable/unrealistic land use change effects seen in this study. However, current data products of historic land use change are often themselves associated with high uncertainty in historic trends, due to data limitations. In order to build improved historical products, merging multiple data sources could support the evaluation of future projected LULC changes. In addition, a clear declaration of potentially possible regional or global LULC pathways in comparison to LULC changes going beyond historical exemplars would be desirable."

*I am still confused about the procedure by which crops are prescribed in LPJ. The total crop fraction evolves according to the given land use scenario, I assume, while the particular mixture of crop types is prescribed to be constant in time after 2006 (table S1). If this mixture is constant in time, then how was it possible to represent crop adaptation by "simulating the adequate selection of suitable crop varieties under changing climate" (section 2.2)?*

The mixture of crop types in terms of C3 crops vs. C4 crops vs. rice (respectively 4 crop types in IMAGE/MAgPIE simulations) follows the MIRCA2000 dataset in the version of Fader et al. (2010) with increasing amount of irrigated crops until 2006. After 2006 crop fractions do not change. This is stated in Table S1. The adaptation of crop types under changing climate, i.e. the dynamic calculation of PHU,

is indeed meant to represent adaptive improvement (or selection) of crop *varieties* under climate change. It does not imply different crop *species*. As already stated in section2.2 "Adaptation to climate change is partially accounted for by a dynamic calculation of potential heat units (PHU) needed for the full development of a crop before harvest, simulating the adequate selection of suitable crop varieties under changing climate (see Lindeskog et al., 2013)".  We suggest to add in section 2.4 additional text such as "Yields respond to changes in climate and $CO_2$, including also adaptation as arising from the calculation of dynamic PHU (see 2.2). Adaptation related to e.g. choosing different crops species in a gridcell was not considered here in simulations of the future period."

*Could you please clarify if NPP is aggregated over all ecosystem types including crops? I suppose this is the case, which would imply that there is some information overlap between the NPP and crop productivity indicators. Would it be possible to show some disaggregated results for NPP (i.e. separately for the 3 types of vegetation represented in LPJ)? It might help to reveal some ecosystem-specific responses.*

Yes, NPP is aggregated over all plant and crop functional types and therefore there is an information overlap between NPP and the crop production indicator. NPP could in principle be split into NPP from crops and other plants. However, from the ecosystem service point of view, NPP as the entirety of the biomass produced in the ecosystem is relevant and not so much the crop/plant types that it comes from. A detailed evaluation of this would be besides the main topic of our study. Therefore, we would not want to distract the reader further by splitting up NPP and with this adding complexity to tables and figures. Also by showing crop production explicitly, an indication is given, where NPP is significantly influenced by crop NPP. We will clarify the definition of NPP and the information overlap with crop production in sect 2.4.

*L31-33: please provide an uncertainty range for all variables along with the median value.*

Will be done by including the means from section 3.2: "The variability in ecosystem service indicators across scenarios was especially high for vegetation carbon stocks (9.2% ± 4.1%) and crop production (31.2% ± 12.2%)."

*L229: Could you clarify how C storage from CCS was quantified, given that BECCS is not represented in LPJ (table S1)?*

Following Krause et al 2017, we assumed 80% of the harvested C from bioenergy crops to be captured and stored. This may be optimistic, but was similar to the assumptions in MagPIE and IMAGE. We will add this to the revised text in section 2.3.

"… LUMs, with cropland also including bioenergy areas and pasture including degraded forests rangeland and grazing land. Following Krause et al. (2017) we assumed in the BECCS scenarios 80% of the harvested C from bioenergy crops to be captured and stored."

*L244: it would be nice to include table S2 in the main text.*

This could certainly be done, and we would change table references in the main manuscript then accordingly (Table S2 would become Table 2).

*L256-260: this part would fit better in the method section. Moreover, it would be good to include a cross-walking table in the method section to explain how the land classes of the respective LUMs/IAMs were translated into the 3 main types in LPJ. i.e., this needs to be clarified not only for CLUMondo.*

Good point, we will integrate these into methods and suggest to add the following table on how LUM scenario classes were translated to the LPJ-GUESS land cover classes to the SI.

Table S2. Translation of LUM land use information to three LPJ-GUESS land use types.

| LPJ-GUESS landcover / LUM landcover | cropland | pasture | natural |
|---|---|---|---|
| CLU-Mondo | regionally varying composition of each CLUMondo land use system in natural, pasture and cropland area following Eitelberg et al., 2016 | | |
| IMAGE | cropland | pasture | forest, urban, other natural |
| MAgPIE | cropland (irrigated, non-irrigated) | pasture | forest, urban, other natural |
| LUH1 | cropland (inclusing bioenergy cropland for 26BE scenario) | pasture | primary vegetation, secondary vegetation, urban |
| LUH2 | C3/C4 annuals, C3/C4 perennial, C3 nitrogen fixing | managed pasture, rangeland | primary land, secondary land, urban |

*Fig 3: please add the units for all variables.*

Fig. 3 shows the change and uncertainty in ES indicators in % as is stated in the figure caption. Adding "[% change]" behind each ES indicator name would crowd the Figure with a lot of txt. However, we suggest to place the unit to a more prominent position in the revised caption to better highlight it.

---

## Author Comment (AC2) · 2 Dec 2020

**Reply to anonymous referee #2**

General comments

*Bayer et al. presented a comparison of multiple land use and land cover (LULC) scenarios broadly used for understanding the future impacts of global change on carbon emission and ecosystem functioning. They highlight the large discrepancies amongst scenarios and what are the implications of this broad variety of scenarios on the trends of several ecosystem service variables. LULC is unquestionably an important driver of future changes and understanding and assessing the uncertainty and variability of such scenarios is critical for both science and policy makers. Therefore, the topic of this study is relevant and timely, and this work can be an important contribution, but it still requires some clarifications in the methodological approach and the interpretation of results, and potentially one additional analysis that should be straightforward to implement and could make the manuscript more appealing.*

We thank the reviewer for these encouraging statements and in the revised version of the manuscript we will address the open points as described below.

*My main concern with the current presentation is that the authors highlight the large discrepancies amongst LULC scenarios and even mention that some scenarios predict unrealistic regional patterns of LULC change. However, I understood that the authors used only one scenario (RCP2.6) from one model (IPSL-CM5A-LR), whereas many of the LCLU scenarios were aligned to a broad range of future scenarios, and potentially had very different initial states that may explain at least some of the discrepancies. Couldn't these differences explain some or most of the discrepancies amongst scenarios? Also, the discrepancy across LULC is not necessarily a bad thing, as many of the scenarios selected by the authors are for different socio-economic pathways and they should be different. The authors could clarify these points in a revised version.*

See our response to Reviewer 1, whose question points into a somewhat similar direction. We agree that the range in LULC scenarios isn't a bad thing at all - this wasn't the message we would like to get across (and we will make this point explicit in the revised version of the manuscript). Nevertheless, even given different initial states and different socio-economic futures, one would nevertheless assess simulated changes in land use from a "common sense" perspective, and critically reflect on some rates and/or directions of change that seem implausible. This would include, for instance, cropland expansion along the fringes of the Sahara or extremely large and rapid area conversions in some countries (e.g., Malawi, Tanzania, Zimbabwe, etc.). These seem quite extreme w.r.t. past trends. Likewise, a rapid reforestation of the Cerrado regions would be a complete reversal of past trends. While it is not completely impossible, of course, we argue that the speed and magnitude of changes in some scenarios would nonetheless seem implausible.

Regarding the selection of our climatic pathway and model: As stated in the methods section, IPSL-CM5A-LR projects future temperature and precipitation changes that lie 'in the middle' of an ensemble of a wider range of GCMs used in the ISI-MIP intercomparison project, which is the reason why we selected this GCM. The variability in some ES indicators when using climate forcing from multiple GCMs was explored in Ahlström et al. (2012) and Schaphoff et al. (2006), as highlighted in section 4.3. In addition, in section 4.2 we presented a summary of the results of a sensitivity test using climate inputs from five GCMs, instead of just IPSL-CM5A-LR, along with the four diverse scenarios from the LUH1 product in our simulation set-up.

*Also, the authors presented and discussed the variability in the trajectory of ecosystem services (ES) across all scenarios. This is fine but I think an analysis comparing the emergent responses across all the scenarios analyzed could give much more insight on how LCLU change could affect the future of ecosystem functioning. For example, the authors could relate the changes in cropland/forest/pasture area in each continent with changes in runoff or evapotranspiration and use the slope of these relationships to understand the sensitivity of ecosystem services to LCLU changes.*

We thank the reviewer for this constructive suggestion. We had indeed tested a similar analysis already when writing the submitted version of the manuscript. In response to the reviewer request, we now performed a direct correlation of the changes ES with the changes in LULC classes, on the scale of biomes. The slopes of these relationships indicate some relatively pronounced responses such as for the increase in vegetation C upon the increase in the natural land fraction, and the increase in crop production upon the increase in cropland fraction – which in fact would be expected. Slopes for other ES indicator responses are mostly low, with little statistical power: $R^2$ values indicating the strength of the relationships are low to very low (majority below 0.1, best $R^2$ is 0.32). This reflects that direct correlations of ES indicator changes with changes in LULC are difficult to establish because they are significantly impaired by, e.g., overlying climate effects, different base levels in ES indicators and LULC configurations and different ecosystem responses below the level of biomes (including, e.g., legacy effects of past LULC changes). Nonetheless we believe it is worthwhile to include these findings in a revised version of the manuscript and address them in the following paragraph that will be added to section 4.2, along with supporting table and figure in the SI.

"A direct correlation of the per cell changes in ES indicators with the corresponding changes in cropland, pasture and natural land fraction could reveal the sensitivity of different ES indicators to changes in LULC. Across all scenarios and for the considered biomes, these relationships suggested for instance an about 1.5 % increase in Vegetation C per percent increase in natural land fraction and between 12 and 24 % increase in crop production per percent increase in the cropland fraction (see Table S8). Emergent responses of other ES indicators to changes in LULC fractions were mostly low (slope of regression lines close to 0) across the biomes (see Table S8). None of the identified relationships provided high reliability (highest $R^2$ was 0.32 for the change in vegetation C per change in natural land fraction in tropical forests, see Fig. S5). This reflects that direct correlations of ES indicator changes with changes in LULC are difficult to establish because they are significantly impaired by, e.g., overlying climate effects, different base levels in ES indicators and LULC configurations and different ecosystem responses below the level of biomes (including, e.g., legacy effects of past LULC changes)."

The following figure shows the direct correlation of the per cell changes in ES indicators with the corresponding changes in cropland, pasture and natural land fraction across all scenarios as an example for the tropical forest biome. It will be added to the SI of the new manuscript along with a table summarizing the slope values for all biomes.

[Figure]

Specific/Minor comments

*Line 29. I don't think abrupt transitions necessarily indicate problems in this case. For example, a tropical forest may be minimally disturbed until new infrastructure (e.g., paved roads) is built. Likewise, policies change with governments and can result in significant reversals of trends, for example, the significant reduction in deforestation in Brazil in the 2000s and the current increase.*

We believe this comment refers mainly to text in lines 464? The reviewer is correct of course that such abrupt transitions indeed do occur. However, past changes in a countries' land use policy are not included in any of the LUM model set up. Thus the modelled rapid transitions unfortunately do not represent a realistic policy process.

*Line 35. I suggest to replace "edge" with boundaries or ecotones. Forest edge is commonly use in landscape scale to define transitions from deforested and forest areas at landscape level (sensu Skole and Tucker 1993). Same comment for Line 75.*

Correct, boundaries is the better word.

*Line 80. Rephrase: large uncertainties in LCLU projections may affect the confidence in projected changes in ecosystem functioning, but not the ecosystem functioning per se.*

You are right, text will be adapted as suggested.

*Line 115. Some word is missing after "following".*

Thanks, sentence could be revised as "summarized in this section and …".

*Section 2.2. Maybe I missed it, but what happens when the LCLU scenarios are inconsistent with the LPJ vegetation? For example, if the scenario indicates timber harvesting but LPJ does not predict any forest? In fact, it is unclear to be whether or not logging was considered in these simulations.*

These are important points - and indeed, wood harvest was not considered in the simulations because the underlying assumptions and technical implementations of harvest in scenarios vary a lot and could produce exactly the complications mentioned here. We will clarify this aspect in the methods section. Of each LULC scenario we only used the fraction of natural, pasture and cropland, as it is described in the methods section. Since harvest is not accounted for in the simulations, other inconsistencies are excluded: only the changes in the fractions of natural, pasture and cropland were considered as a change from the baseline LULC, i.e. they are consistent with the baseline (e.g. fraction of natural cell can only be reduced by the amount that is existent in e.g., year 2000 in the scenario period).

*Line 195. What was the rationale for selecting RCP2.6 instead of other pathways? It seems to me that RCP2.6 is rather too optimistic.*

We agree that given the still very high annual $CO_2$ emissions, RCP2.6 seems a rather unrealistic pathway. Our chief reason to select RCP2.6 was to keep the impact of climate on projected changes and uncertainties in ES indicators relatively low, and to focus chiefly on the land-use change impacts. This will be included in the revised manuscript in section 2.3, for instance along the lines of "Climate projections followed the RCP 2.6 pathway. Large degrees of climate change and high atmospheric

CO2 concentrations can have large impacts on ecosystem service indicators (see, e.g., Alexander et al., 2018). As our focus here is on the impact of land-use change, we chose a climate change projection, which over the simulation period would have relatively little additional impact".

*Line 209. Related to my point in section 2.2. It is fine to group all the natural classes, but this still does not clarify what happens in the case of logging.*

As stated above, the revised manuscript will specify that harvest and logging was not explicitly simulated. In addition, we added a table to the SI that explicitly shows the translation of LUMs LULC classes to the LPJ-GUESS LULC classes cropland, pasture and natural.

*Line 231. I understand the rationale for minimizing the role of interannual variability, but it is unclear to me that 5 years is sufficient. Would the results change considerably if, for example, 10-year averages were used instead?*

In our experience this longer averaging period would not change the observed trends in ES indicators. We chose here a 5-year interval to "dissect" the relatively short simulation period into a meaningful number of sub-periods.

*Section 3. The authors have a separate discussion section, but the text in the results section often read more like discussion (e.g., most of the paragraph starting in line 391). Also, I think a multi-panel figure that showed the average changes in cropland, pasture, and forest (both increase and decrease) would help to summarize the results.*

This is a valid observation. In some sections of the results section we include reference to literature, which could be interpreted as a discussion. Specifically, when we refer to underlying processes that would explain a simulated change in ecosystems. We did this in order to focus the discussion specifically on the land-use change impacts and uncertainties. We acknowledge that this blurs to some degree the clean boundaries between results and discussion, but in our view concentrates the discussion on the main aspects of the paper - while still not omitting the links of our findings to the existing literature.

Figure: we discussed such a figure averaging across all scenarios for the original submission. However, since we are examining here scenarios that have very diverging underlying socio-economic assumption, we think it would be erroneous to produce an average across all these, which has the danger to leave the reader with the impression that such a figure would represent some form of an ensemble mean. Instead, Fig. 2 provides a summary of the dominant LULC changes for each scenario separately.

*Line 273. This sentence is confusing.*

Thanks, well spotted: We will revise the sentence along the lines of "In IMAGE, food production meeting underlying societal demand has large priority. "

*Line 288. Examples of some countries?*

*This refers to the countries listed at the beginning of the paragraph and the areas marked in red in fig. 2 showing quite radical increases in the cropland fraction. We will adapt the sentence to clarify which countries are meant. "Here, some countries seem to provide substantially cheaper commodity*

*prices than others, explaining the radical changes seen in the regions as listed above (compare also Fig. 2)."*

*290 changes seen in some countries or regions.*

*see previous comment*

*Line 372. "Central" instead of "Middle"?*

Thanks, will be corrected.

*Lines 391–422. Can changes in irrigation also contribute to changes in ET and runoff in LPJ-GUESS? Does LPJ-GUESS simulate irrigation?*

As a stand-alone model as used in this study, LPJ-GUESS simulates irrigation in a simplified way, using the prescribed fractions of rainfed vs. irrigated management for each crop type, which are identical across all simulations (see methods section). Changes in ET and runoff are therefore consequences solely from the LULC change. Of course, e.g. an increase in cropland in a cell with high irrigated fraction and therefore increase crop growth will have higher effects on ET and runoff.

Only when LPJ-GUESS is bi-directionally coupled to a socio-economic land-use change model (such as in e.g., Alexander et al., Global Change Biology 2018, doi: 10.1111/gcb.14110), LPJ-GUESS adapts irrigation patterns to demands. We will clarify in the revised manuscript.

*Line 464. It was more than soy moratorium in the case of Brazil, law enforcement and policy changes were also important (Nepstad et al. 2014).*

We will clarify this in the revised manuscript.

*Line 638. I suggest "South American" Cerrado and Chaco instead to remove ambiguity, as the Chaco is not in Brazil.*

Well spotted, thanks - this will be corrected.

*Line 709. I agree with this paragraph and this is why I also suggested the analysis on the emergent responses. It seems that the authors already have the results ready for at least some initial analysis to qualify the changes in ES responses as functions of LULC changes.*

We appreciate your positive feedback. The suggested analysis was included as stated above.

*Figures 1 and 4. Some of the colours are difficult to distinguish, at least for me (CLU and LUH2, for example in Fig. 1). For Figure 4, the authors could fix hues for different levels of relative change (rows), and fix brightness for the absolute values (columns), it would also make the figure more intuitive.*

Thanks, we will ensure that the colors in the Figures are easier to read. We revised Fig. 4 and increased the brightness for the less important categories in order to make the figure more intuitive and also changed the color for CLU in Fig. 1.

References

Nepstad, D., D. McGrath, C. Stickler, A. Alencar, A. Azevedo, B. Swette, T. Bezerra, M. DiGiano, J. Shimada, R. Seroa da Motta, E. Armijo, L. Castello, P. Brando, M. C. Hansen, M. McGrath-Horn, O. Carvalho, and L. Hess, 2014: Slowing Amazon deforestation through public policy and interventions in beef and soy supply chains. Science, 344 (6188), 1118–1123, doi:10.1126/science.1248525.

Skole, D., and C. Tucker, 1993: Tropical deforestation and habitat fragmentation in the Amazon: Satellite data from 1978 to 1988. Science, 260 (5116), 1905–1910, doi:10.1126/science.260.5116.1905.

---

## Author Response (AR1)

Dear Editor,

thanks for allowing us to submit a revised version of the manuscript. We briefly describe below the changes done - which follow closely the comments we had posted on the Discussion site.

Changes in the revised manuscript are marked as track changes to facilitate their easy finding by you and the reviewers.

Yours sincerely,

Anita Bayer, Almut Arneth et al.

**Reply to anonymous referee #1**

This study analyzes an ensemble of future land use projections arising from a set of 5 LUMs/IAMs. These land use projections are then translated into changes in ecosystem services (ES) using the LPJ-GUESS model. One of the main conclusions is that there is a large spread in land use projections (and therefore ES) and that most of this spread originates from structural model differences (i.e. choice of LUM/IAM) rather than from socio-economic assumptions (i.e. choice of scenario storyline within a given model). Overall, this is an interesting study which makes an important point about the large uncertainties associated with future land use projections and their potential causes. However some aspects of the manuscript could still be improved and require additional clarifications.

We thank the reviewer for the expressed interest in our manuscript. In the revisions to the manuscript we have addressed the raised questions as described below. Line numbers refer to the track changes document.

I found the whole discussion about the role of the baseline level very hard to follow. I understand from the analysis that there are very different starting states in 2000- 2004 across the models. But does it affect the conclusions of the study or not? One could for instance intuitively expect that in a model starting with lower present-day natural vegetation there would more room for future expansion. More specifically, in which way is the baseline level taken into account when looking at relative changes (L236-239)? And where is the assumption that "Effects of differences in the modeling protocol of CLUMondo/LUH1/LUH2 and IMAGE/MAgPIE simulations for projections of ecosystem dynamics affect especially the base level of ES indicators in 2000-2004 and the relative deviations over time, which this study is focused on, only to a small degree" coming from? This last sentence by the way is so convoluted that I might just have misinterpreted its meaning. On that same issue, it is somewhat disturbing that the IPSL-based climate forcing has been bias-corrected, implying a sort of harmonization between observed and projected climate, while no harmonization was performed for the land use forcing. Why not following the same logic for both climate and land use?

Starting with the last question first: the main reason for using a bias corrected climate is not so much the seamless historical-future transition (which is essential for e.g. carbon and water cycle simulations: without a harmonized climate time series, an artificial offset would be created between the end of the historical and the beginning of the projected future climate), but the fact that simulated climate output

from GCMs is much more reliable in terms of anomalies, rather than in terms of absolute values – most GCMs are biased compared to "real" climate values in a grid cell. Therefore, projected raw daily GCM output has to be corrected, using the differences in the mean and variability between GCM and observations in a reference period.

The reviewer is correct that in principle one could also do that for land-use historical-to-future timeseries. However, doing so would to large degree affect the underlying objectives of our study. We aim to show here the large uncertainty/variability in terms of impacts on ecosystems that arises both from the very different (unknown) socio-economic futures as well as from the different LUC modelling approaches. This includes keeping the differences in the historical baseline, although the latter is less prominent in the manuscript by us concentrating on the relative changes. We will clarify this in the manuscript upon revision, and by doing so will also revise the sentence "Effects of...", which is indeed poorly phrased.

Revisions in the revised ms are (lines 216-231): "The differences in the modeling protocol of CLUMondo/LUH1/LUH2 and IMAGE/MAgPIE simulations will affect the base level of ES indicators in 2000-2004 to some degree, although the impacts of slightly diverging historical model periods, spinup and historical climate would have mostly disappeared by the beginning of the 21st century (baseline period). Larger effects arise from the differences in the individual LUMs per se (see also Alexander et al., 2017). In principle, differences in the baseline land-cover maps could spill-over to the simulated degree of change in the future scenarios. For instance, presence or absence of natural vegetation in the baseline maps might translate into variable degrees of future (semi)natural vegetation re-growth. However, this would only be an important consideration when comparing similar scenarios (and their underpinning storylines related to e.g. sustainability). The alternative approach of harmonizing the differences in ES indicators which would be contrary to our objectives. Therefore, LUM data were taken as they are, with each LUM scenario providing a seamless transition from historical to future, which is needed to simulate vegetation and carbon cycle responses."

I have been wondering whether the positive trend in some ES indicators could be affected or even reversed if fires were properly accounted for. Referring to fire and other processes, section 4.2 has a rather elusive statement: "Because these processes are only to some degree implemented in LPJ-GUESS (see, e.g., Pugh et al., 2019), this could further increase the regional variability in ES indicators as indicated in this study." Something more explicit would be welcome, such as clarifying upfront in section 2.2 how fires are accounted for in the model.

LPJ-GUESS, like other DGVMs, models fire explicitly, but has a simplified representation of other forms of disturbances (which can arise from storms, or insect attaches, etc.). We have clarified this both in the methods (lines 199-201), as well as in section 4.2. (lines 728-731):

"As only fire is explicitly simulated as a disturbance process in LPJ-GUESS, while other forms of disturbances are subsumed in a stochastic background-disturbance, this could further increase the regional variability in ES indicators(see, e.g., Pugh et al., 2019)."

The conclusion that "some scenarios show questionable and possibly unrealistic features in their LULC allocations" could warrant some more in-depth evaluation of historical trends to be fully supported. Although this might be an ambitious task, the discussion could at least outline some evaluation strategies that should be deployed in future studies in order to pinpoint more specific deficiencies.

The reviewer is certainly right in that a full evaluation of pros and cons of the implementation of scenarios' storylines is beyond our study. We aspired to put scenarios' projected LULC changes in a historic context because this is our most realistic reference which led us to the presented conclusion. We added the following text to the manuscript in order to take up the suggestions and clarifying on our goal and procedure:

Lines 495-499 (section 4.1, second aspect on Future regional change rates in a historic context, entry statement added): "Even given different initial states in LULC and different socio-economic pathways of the 16 scenarios, we critically assess the spatial patterns, directions and rates of regional change based on past LULC changes. While it is not completely impossible, of course, we argue that a speed and magnitude extremely opposing trends observed in the past seem at least questionable."

Lines 599-606 (section 4.1, aspect 3 on regional LULC patterns): "In this context it seems worthwhile for the land use community to evaluate simulated future land-use changes against historic trends, in spatial, temporal and thematic aspects. This may avoid some of the questionable, possibly unrealistic, land-use change effects seen in this study. However, current data products of historic land-use change are often themselves associated with high uncertainty in historic trends, due to data limitations. Improved historical products that merge multiple data sources could support the evaluation of future projected LULC changes. In addition, a clear declaration of scenarios showing potentially possible regional or global LULC pathways in comparison to those showing LULC changes going beyond historical exemplars would be desirable."

I am still confused about the procedure by which crops are prescribed in LPJ. The total crop fraction evolves according to the given land use scenario, I assume, while the particular mixture of crop types is prescribed to be constant in time after 2006 (table S1). If this mixture is constant in time, then how was it possible to represent crop adaptation by "simulating the adequate selection of suitable crop varieties under changing climate" (section 2.2)?

The mixture of crop types in terms of C3 crops vs. C4 crops vs. rice (respectively 4 crop types in IMAGE/MAgPIE simulations) follows the MIRCA2000 dataset in the version of Fader et al. (2010) with increasing amount of irrigated crops until 2006. After 2006 crop fractions do not change. This is stated in Table S1. The adaptation of crop types under changing climate, i.e. the dynamic calculation of PHU, is indeed meant to represent adaptive improvement (or selection) of crop *varieties* under climate change. It does not imply different crop *species*. As already stated in section 2.2 "Adaptation to climate change is partially accounted for by a dynamic calculation of potential heat units (PHU) needed for the full development of a crop before harvest, simulating the adequate selection of suitable crop varieties under changing climate (see Lindeskog et al., 2013)".

We add in section 2.4 as further explanation (lines 278-281) "Yields respond to changes in climate and CO2, including also some degree of adaptation, which arises from the calculation of dynamic PHU (see 2.2). Adaptation related to, e.g., choosing different crop species in a grid-cell was not considered here in simulations of the future period."

Could you please clarify if NPP is aggregated over all ecosystem types including crops? I suppose this is the case, which would imply that there is some information overlap between the NPP and crop productivity indicators. Would it be possible to show some disaggregated results for NPP (i.e. separately for the 3 types of vegetation represented in LPJ)? It might help to reveal some ecosystem-specific responses.

Yes, NPP is aggregated over all plant and crop functional types and therefore there is an information overlap between NPP and the crop production indicator. NPP could in principle be split into NPP from crops and other plants. However, from the ecosystem service point of view, NPP as the entirety of the biomass produced in the ecosystem is relevant and not so much the crop/plant types that it comes from. A detailed evaluation of this would be besides the main topic of our study. Therefore, we would not want to distract the reader further by splitting up NPP and with this adding complexity to tables and figures. Also by showing crop production explicitly, an indication is given, where NPP is significantly influenced by crop NPP.

We have clarified this in sect 2.4. (lines 275-276). "All plant and crop functional types contribute to an ecosystems' NPP. Therefore, NPP and crop production are positively correlated."

**L31-33: please provide an uncertainty range for all variables along with the median value.**

We included in the abstract the means from section 3.2 (line 33): "The variability in ecosystem service indicators across scenarios was especially high for vegetation carbon stocks (9.2%  $\pm$  4.1%) and crop production (31.2%  $\pm$  12.2%)."

**L229: Could you clarify how C storage from CCS was quantified, given that BECCS is not represented in LPJ (table S1)?**

Following Krause et al 2017, we assumed 80% of the harvested C from bioenergy crops to be captured and stored. This may be optimistic, but was similar to the assumptions in MagPIE and IMAGE. We added accordingly to the revised text in section 2.3:

Line 246: "Cropland fractions also included bioenergy areas..."

Lines 257-258: "In the BECCS scenarios we assumed 80% of the harvested C from bioenergy crops to be captured and stored following Krause et al (2017)."

**L244: it would be nice to include table S2 in the main text.**

Done, and we also changed table references in the main manuscript accordingly (Table S2 is now Table 2).

L256-260: this part would fit better in the method section. Moreover, it would be good to include a cross-walking table in the method section to explain how the land classes of the respective LUMs/IAMs were translated into the 3 main types in LPJ. i.e., this needs to be clarified not only for CLUMondo.

Good point, we integrate these into methods and added the following table on how LUM scenario classes were translated to the LPJ-GUESS land cover classes to the SI.

**Table S2. Translation of LUM land use information to three LPJ-GUESS land use types.**

| LPJ-GUESS landcover | cropland                                                                            | pasture | natural |
|---------------------|-------------------------------------------------------------------------------------|---------|---------|
|                     |                                                                                     |         |         |
|                     |                                                                                     |         |         |
| ▼ LUM landcover     |                                                                                     |         |         |
| CLU-Mondo           | regionally varying composition of each CLUMondo land use system in natural, pasture |         |         |
|                     | and cropland area following Eitelberg et al., 2016                                  |         |         |

| IMAGE  | cropland (irrigated, non- | pasture, degraded forests | forest, urban, other    |
|--------|---------------------------|---------------------------|-------------------------|
|        | irrigated)                |                           | natural                 |
| MAgPIE | cropland (irrigated, non- | pasture                   | forest, urban, other    |
|        | irrigated)                |                           | natural                 |
| LUH1   | cropland (inclusing       | pasture                   | primary vegetation,     |
|        | bioenergy cropland for    |                           | secondary vegetation,   |
|        | 26BE scenario)            |                           | urban                   |
| LUH2   | C3/C4 annuals, C3/C4      | managed pasture,          | primary land, secondary |
|        | perennial, C3 nitrogen    | rangeland                 | land, urban             |
|        | fixing                    |                           |                         |

**Fig 3: please add the units for all variables.**

Fig. 3 shows the change and uncertainty in ES indicators in % as is stated in the figure caption. Adding "[% change]" behind each ES indicator name would crowd the Figure with a lot of text. However, we now placed the unit to a more prominent position in the revised caption to better highlight it.

**Reply to anonymous referee #2**

**General comments**

Bayer et al. presented a comparison of multiple land use and land cover (LULC) scenarios broadly used for understanding the future impacts of global change on carbon emission and ecosystem functioning. They highlight the large discrepancies amongst scenarios and what are the implications of this broad variety of scenarios on the trends of several ecosystem service variables. LULC is unquestionably an important driver of future changes and understanding and assessing the uncertainty and variability of such scenarios is critical for both science and policy makers. Therefore, the topic of this study is relevant and timely, and this work can be an important contribution, but it still requires some clarifications in the methodological approach and the interpretation of results, and potentially one additional analysis that should be straightforward to implement and could make the manuscript more appealing.

We thank the reviewer for these encouraging statements and in the revised version of the manuscript addressed the open points as described below. Line numbers refer to the track changes document.

My main concern with the current presentation is that the authors highlight the large discrepancies amongst LULC scenarios and even mention that some scenarios predict unrealistic regional patterns of LULC change. However, I understood that the authors used only one scenario (RCP2.6) from one model (IPSL-CM5A-LR), whereas many of the LCLU scenarios were aligned to a broad range of future scenarios, and potentially had very different initial states that may explain at least some of the discrepancies. Couldn't these differences explain some or most of the discrepancies amongst scenarios? Also, the discrepancy across LULC is not necessarily a bad thing, as many of the scenarios selected by the authors are for different socio-economic pathways and they should be different. The authors could clarify these points in a revised version.

See our response to Reviewer 1, whose question points into a somewhat similar direction. We agree that the range in LULC scenarios isn't necessarily a bad thing - this wasn't the message we wanted to get across. We make this point more explicit in the revised version of the manuscript (lines 95-100): "We intend to also highlight how different LULC patterns impact ecosystems and related ES indicators. This supports the interpretation of conclusions derived from LUMs and IAMs towards policy decisions for instance on intensification, conservation or climate change mitigation options. A broad range of future LULC projections from different LUMs and different socio-economic assumptions is important, given the unknown future. Nevertheless, assessing critically the spatial pattern and rates of change can support their interpretation in terms of plausibility."

Still, even given different initial states and different socio-economic futures, one would assess simulated changes in land use from a "common sense" perspective, and critically reflect on some rates and/or directions of change that seem implausible. This would include, for instance, cropland expansion along the fringes of the Sahara or extremely large and rapid area conversions in some countries (e.g., Malawi, Tansania, Zimbabwe, etc.). These seem quite extreme w.r.t. past trends. Likewise, a rapid reforestation of the Cerrado regions would be a complete reversal of past trends. While it is not completely impossible, of course, we argue that the speed and magnitude of changes in some scenarios would nonetheless seem implausible. We added an entry statement on this at the beginning of the second argument of section 4.1 (lines 495-499).

Regarding the selection of our climatic pathway and model: As stated in the methods section, IPSL-CM5A-LR projects future temperature and precipitation changes that lie 'in the middle' of an ensemble of a wider range of GCMs used in the ISI-MIP intercomparison project, which is the reason why we selected this GCM. The variability in some ES indicators when using climate forcing from multiple GCMs was explored in Ahlström et al. (2012) and Schaphoff et al. (2006), as highlighted in section 4.3 (lines 765-767). In addition, in section 4.2 we presented a summary of the results of a sensitivity test using climate inputs from five GCMs, instead of just IPSL-CM5A-LR, along with the four diverse scenarios from the LUH1 product in our simulation set-up (lines 667-671).

Also, the authors presented and discussed the variability in the trajectory of ecosystem services (ES) across all scenarios. This is fine but I think an analysis comparing the emergent responses across all the scenarios analyzed could give much more insight on how LCLU change could affect the future of ecosystem functioning. For example, the authors could relate the changes in cropland/forest/pasture area in each continent with changes in runoff or evapotranspiration and use the slope of these relationships to understand the sensitivity of ecosystem services to LCLU changes.

We thank the reviewer for this constructive suggestion. We had indeed tested a similar analysis already when writing the submitted version of the manuscript. In response to the reviewer request, we now performed a direct correlation of the changes ES with the changes in LULC classes, on the scale of biomes. The slopes of these relationships indicate some relatively pronounced responses such as for the increase in vegetation C upon the increase in the natural land fraction, and the increase in crop production upon the increase in cropland fraction – which in fact would be expected.

Slopes for other ES indicator responses are mostly low, with little statistical power: R2 values indicating the strength of the relationships are low to very low (majority below 0.1, best R2 is 0.32). This reflects that direct correlations of ES indicator changes with changes in LULC are difficult to establish because they are significantly impaired by, e.g., overlying climate effects, different base levels in ES indicators and LULC configurations and different ecosystem responses below the level of biomes (including, e.g., legacy effects of past LULC changes). Nonetheless we believe it is worthwhile to include these findings in a revised version of the manuscript and address them in the following paragraph added to section 4.2 (lines 737-749), along with supporting Table S8 and Figure S5 in the SI; an example of this had already been included in our comment on the Discussion page.

"A direct correlation of the per grid-cell changes in ES indicators with the corresponding changes in cropland, pasture and natural land fraction could reveal the sensitivity of different ES indicators to changes in LULC. Across all scenarios and for the considered biomes, these relationships suggested for instance an about 1.5% increase in vegetation C per percent increase in natural land fraction and between 12 and 24% increase in crop production per percent increase in the cropland fraction (see Table S8). Emergent responses of other ES indicators to changes in LULC fractions were mostly low (slope of regression lines close to 0) across the biomes (see Table S8). None of the identified relationships provided high reliability (highest R2 was 0.32 for the change in vegetation C per change

in natural land fraction in tropical forests, see Fig. S5). This reflects that direct correlations of ES indicator changes with changes in LULC are difficult to establish because they are significantly impaired by, e.g., overlying climate effects, different base levels in ES indicators and LULC configurations and different ecosystem responses below the level of biomes (including, e.g., legacy effects of past LULC changes)."

**Specific/Minor comments**

Line 29. I don't think abrupt transitions necessarily indicate problems in this case. For example, a tropical forest may be minimally disturbed until new infrastructure (e.g., paved roads) is built. Likewise, policies change with governments and can result in significant reversals of trends, for example, the significant reduction in deforestation in Brazil in the 2000s and the current increase.

We believe this comment refers mainly to text in lines 464? The reviewer is correct of course that such abrupt transitions indeed do occur. However, past changes in a countries' land use policy are not included in any of the LUM model set up. Thus the modelled rapid transitions unfortunately do not represent a realistic policy process.

Line 35. I suggest to replace "edge" with boundaries or ecotones. Forest edge is commonly use in landscape scale to define transitions from deforested and forest areas at landscape level (sensu Skole and Tucker 1993). Same comment for Line 75.

Correct, boundaries is the better word, has been corrected.

*Line 80. Rephrase: large uncertainties in LCLU projections may affect the confidence in projected changes in ecosystem functioning, but not the ecosystem functioning per se.*

You are right, text adapted as suggested (line 80).

Line 115. Some word is missing after "following".

Thanks, sentence was revised as "summarized in this section and ..." (line 123).

Section 2.2. Maybe I missed it, but what happens when the LCLU scenarios are inconsistent with the LPJ vegetation? For example, if the scenario indicates timber harvesting but LPJ does not predict any forest? In fact, it is unclear to be whether or not logging was considered in these simulations.

These are important points - and indeed, wood harvest was not considered in the simulations because the underlying assumptions and technical implementations of harvest in scenarios vary a lot and could produce exactly the complications mentioned here. We will clarify this aspect in the methods section. Since harvest is not accounted for in the simulations (line 256-257), other inconsistencies are excluded: only the changes in the fractions of natural, pasture and cropland were considered as a change from the baseline LULC (compare databases for historical land use underlying each scenario, see Table S1), i.e. they are consistent with their respective baseline (e.g. fraction of natural cell can only be reduced by the amount that is existent in e.g., year 2000 in the scenario period). *Line 195. What was the rationale for selecting RCP2.6 instead of other pathways? It seems to me that RCP2.6 is rather too optimistic.*

We agree that given the still very high annual  $CO_2$  emissions, RCP2.6 seems a rather unrealistic pathway. Our chief reason to select RCP2.6 was to keep the impact of climate on projected changes and uncertainties in ES indicators relatively low, and to focus chiefly on the land-use change impacts. This is now included in the revised manuscript in section 2.3 (lines 208-212):

"Climate projections and  $CO_2$  concentrations followed the RCP 2.6 pathway. Large magnitudes of climate change and high atmospheric  $CO_2$  concentrations affect ES indicators notably (see, e.g., Alexander et al., 2018). As our focus here is on the impact of land-use change, we chose a climate change projection, which over the simulation period would have relatively little additional impact."

*Line 209. Related to my point in section 2.2. It is fine to group all the natural classes, but this still does not clarify what happens in the case of logging.*

As stated above, the revised manuscript we specified that harvest and logging was not explicitly simulated (lines 256-257). In addition, we added Table S2 to the SI that explicitly shows the translation of LUMs LULC classes to the LPJ-GUESS LULC classes cropland, pasture and natural.

Line 231. I understand the rationale for minimizing the role of interannual variability, but it is unclear to me that 5 years is sufficient. Would the results change considerably if, for example, 10-year averages were used instead?

In our experience this longer averaging period would not change the observed trends in ES indicators. We chose here a 5-year interval to "dissect" the relatively short simulation period into a meaningful number of sub-periods.

Section 3. The authors have a separate discussion section, but the text in the results section often read more like discussion (e.g., most of the paragraph starting in line 391). Also, I think a multi-panel figure that showed the average changes in cropland, pasture, and forest (both increase and decrease) would help to summarize the results.

This is a valid observation. In some sections of the results section we include reference to literature, which could be interpreted as a discussion. Specifically, when we refer to underlying processes that would explain a simulated change in ecosystems. We did this in order to focus the discussion specifically on the land-use change impacts and uncertainties. We acknowledge that this blurs to some degree the clean boundaries between results and discussion, but in our view concentrates the discussion on the main aspects of the paper - while still not omitting the links of our findings to the existing literature.

Figure: we discussed such a figure averaging across all scenarios for the original submission. However, since we are examining here scenarios that have very diverging underlying socio-economic assumption, we think it would be erroneous to produce an average across all these, which has the danger to leave the reader with the impression that such a figure would represent some form of an ensemble mean. Instead, Fig. 2 provides a summary of the dominant LULC changes for each scenario separately.

Line 273. This sentence is confusing.

Thanks, well spotted: We revised the sentence "In IMAGE, food production meeting the underlying societal demand has large priority." (lines 318-319)

**Line 288. Examples of some countries?**

This refers to the countries listed at the beginning of the paragraph and the areas marked in red in fig. 2 showing quite radical increases in the cropland fraction. We adapted the sentence to clarify which countries are meant (line 336). "... the radical changes seen in the regions as listed above (compare also Fig. 2)."

**290 changes seen in some countries or regions.**

see previous comment.

**Line 372. "Central" instead of "Middle"?**

Thanks, corrected.

**Lines 391–422. Can changes in irrigation also contribute to changes in ET and runoff in LPJ-GUESS? Does LPJ-GUESS simulate irrigation?**

As a stand-alone model as used in this study, LPJ-GUESS simulates irrigation in a simplified way, using the prescribed fractions of rainfed and irrigated management for each crop type (see methods section). Changes in ET and runoff are therefore consequences from the LULC change in combination with changes in crop type fractions (IMAGE and MAgPIE simulations use CFT information from IAMs, all other simulations use the same CFT information, see Table S1). Of course, e.g. an increase in cropland in a cell with high irrigated fraction and therefore increased crop growth will have higher effects on ET and runoff. We inserted a statement here, that changes in irrigation pattern may contribute to changes in runoff and ET:

Lines 446-447: "Also changes in irrigation patterns affect water runoff (IMAGE and MAgPIE simulations only, see methods)."

Only when LPJ-GUESS is bi-directionally coupled to a socio-economic land-use change model (such as in e.g., Alexander et al., Global Change Biology 2018, doi: 10.1111/gcb.14110), LPJ-GUESS adapts irrigation patterns to demands.

**Line 464. It was more than soy moratorium in the case of Brazil, law enforcement and policy changes were also important (Nepstad et al. 2014).**

**Thanks, clarified in the revised manuscript.**

Lines 516-519: "Indeed, some rapid land-use changes have occurred in the past, caused by unexpected disruptions in markets or governance structures (e.g., Brazil's soy moratorium combined with enforcement of related policies, e.g., Nepstad et al., 2014; Gibbs et al., 2015; collapse of the Soviet union, e.g. Hostert et al., 2011)."

*Line 638. I suggest "South American" Cerrado and Chaco instead to remove ambiguity, as the Chaco is not in Brazil.*

Well spotted, thanks - corrected.

Line 709. I agree with this paragraph and this is why I also suggested the analysis on the emergent responses. It seems that the authors already have the results ready for at least some initial analysis to qualify the changes in ES responses as functions of LULC changes.

We appreciate your positive feedback. The suggested analysis was included as stated above.

Figures 1 and 4. Some of the colours are difficult to distinguish, at least for me (CLU and LUH2, for example in Fig. 1). For Figure 4, the authors could fix hues for different levels of relative change (rows), and fix brightness for the absolute values (columns), it would also make the figure more intuitive.

Thanks, good point. We revised Fig. 4 and increased the brightness for the less important categories in order to make the figure more intuitive and also changed the colour for CLU in Fig. 1.

**References**

Nepstad, D., D. McGrath, C. Stickler, A. Alencar, A. Azevedo, B. Swette, T. Bezerra, M. DiGiano, J. Shimada, R. Seroa da Motta, E. Armijo, L. Castello, P. Brando, M. C. Hansen, M. McGrath-Horn, O. Carvalho, and L. Hess, 2014: Slowing Amazon deforestation through public policy and interventions in beef and soy supply chains. Science, 344 (6188), 1118–1123, doi:10.1126/science.1248525.

Skole, D., and C. Tucker, 1993: Tropical deforestation and habitat fragmentation in the Amazon: Satellite data from 1978 to 1988. Science, 260 (5116), 1905–1910, doi:10.1126/science.260.5116.1905.

**Reply to short comment by Judith Verstegen**

This study analyzes 16 scenarios from five land use models (LUMs) and integrated assessment models (IAMs) and the effects of the resulting land use changes towards 2040 on ecosystem service (ES) indicator values. The text is easy to read, the work in well-embedded in existing literature, and the results are visualizaed comprehensively. I have two main questions/concerns with respect to this study, as detailed below.

Thanks for the overall positive statement on the manuscript. We answer the questions below & incorporated these into the revised manuscript.

Model-coupling mismatches: To my understanding, you use a one-way coupling between the LUM/IAM and LPJ-GUESS. This can lead to mismatches. For example, the IAM has computed production of particular agricultural products for a region, and converted this into a cropland area per grid cell. In LPJ-GUESS, this grid cell contains a prescribed fraction of crop functional types, which can mismatch with the products in the IAM, and a certain yield, which can mismatch with the production (supply) of agricultural area of that grid cell in the IAM. I believe that such mismatches have a large effect on the results. I expect that large differences in ES indicator values between two models will occur when one of them has a good match with LPJ-GUESS (due to the use of the same data sources) and another a bad match. The comparison between ES indicator values is not fair in this case, because for the second model, its assumptions are violated by the LPJ-GUESS model (for example, demand in the IAM does not match supply computed by LPJ-GUESS). I think it would help to explain how the models are coupled exactly (what variable(s) is/are exchanged) and to bring this point up in your discussion if you agree with this potential issue.

We follow in the manuscript the standard approach in how land use and land cover changes simulated in LUM/IAMs are being used in the ecosystem modelling community (but also in e.g., species distribution models, hydrology models) to assess the impacts on ecosystem processes or biodiversityrelated variables. Implicitly in such an approach is that some of the variables assessed in an ecosystem model would also be computed in the models that deliver the land use change scenarios – most notably crop yields or some carbon-cycle or water-cycle related variables. This is unavoidable when using projections of e.g. land use change from one model type, and using these in another model type. In fact, a similar criticism can be levelled on using climate projections for ecosystem models: the ESM that produces these climate projections would often have a different (and often a more simplistic representation of) vegetation in a grid-cell (and this vegetation would affect the computed climate) compared to e.g. LPJ-GUESS.

However, these inconsistencies would only be an issue if variables computed with LPJ-GUESS would be directly compared to variables computed in the IAMs/LUMs (or: in an ESM). For instance, if we compared computed yields in the different models, or runoff. But this is not our objective here: we take a range of land-use change projections at "face value" and investigate impacts on ecosystem variables in LPJ-GUESS. In principle we could perform a similar experiment with completely stylised or randomised land-use change scenarios that are not computed by a LUM/IAM at all. However, this would make the analysis a purely technical one and would not highlight the large impacts that different socio-economic scenarios have on ecosystems.

Therefore, we don't fully share the expressed concern, but we note this point in the revised manuscript, for clarity (lines 323-237): "Some of the variables assessed in LPJ-GUESS would also be computed in the models that deliver the LULC change scenarios – most notably crop yields and some carbon-cycle or water-cycle related variables. The spatial patterns of these would differ in the LUMs and LPJ-GUESS. However, this does not affect our analysis: here we take the LULC change projections in a uni-directional approach to assess impacts on ecosystem processes; we do not compare similar ecosystem output variables across different model types."

Scenario projections: It is recognized that LUMs/IAMs do not provide predictions (as weather predictions do), but instead projections, meaning that they are conditional (what if ... ?). What conditions are evaluated depends on the question at hand. The conditions can range from realistic (business as usual) to very irrealistic. Irrealistic scenarios can still be useful as thought experiments, to better understand the system, to serve as warnings for worst-case effects, or to evaluation potential policy interventions. As such, in lines 487 and further, you discuss that some scenario results don't seem plausible. But perhaps they aren't meant to be plausible. Therefore, given that the conditions, and thus the scenarios, depend very much on the question asked in the original study, it is not clear to me what you are exactly evaluating when assessing the variation is ES indicator outcomes over all these models and scenarios combined. I could see the added value of analyzing ES indicator outcomes of all business as usual scenarios, as that would show the effect of different assumptions about the working of the current system on ES impacts, but the value of comparing among the other scenarios (which could have easily been very different if a different questions were asked) is not quite clear to me. In the current version of the manuscript, you only comment about this seems to be "However, conclusions drawn here in regard to projected changes in LULC and ES indicators are inherently dependent on the selected set of LUMs and scenarios, evaluation time period and simulation set-up", which does not really help the reader to see what can and cannot be learned from the results given this dependence.

All scenarios analysed here have different socio-economic storylines underpinning them, which are linked to the SSP framework. These storylines indeed represent different, unknown futures. But none of them have been specifically designed to be unrealistic. By contrast, great effort has been put into making each storyline internally consistent (see e.g., O'Neill BC, Kriegler E, Ebi KL, Kemp-Benedict E, Riahi K, et al. 2017. The roads ahead: Narratives for shared socioeconomic pathways describing world futures in the 21st century. Global Environmental Change 42: 169-80). The original studies that led to the various LULC scenarios we investigate here were all designed to project land use change under a plausible future. Our purpose here is to not only compare same style of scenarios (e.g. BAU – which in itself would be a valuable analysis to do, we agree) but to highlight the large impact unknown future land use change has on ecosystems. Most people are well aware that different climate trajectories (RCPs) will greatly affect ecosystems. Even climate change projections for a single RCP when realised with different ESMs will result in large variability in computed ecosystem outcomes. The fact that similarly large variability can be introduced by land-use change (within or between e.g. an SSP) is less known – which is one of our objectives (we will highlight this in the revised version of the manuscript, page 2, see below). At the same time, we also observe that the rate or direction of some of the land use change projections seem implausible, irrespective of the underlying storyline.

**Minor comments:**

481-486: In the context of this paragraph, which speculates about the potentially more valid smallscale changes of some of the models, you may be interested to know that, in a recent study of LUC in Brazil, we found that indeed the small-scale changes of a spatially-explicit LUM were more accurate than the larger-scale changes of an economic model, see Stepanov et al. 2020, doi:10.3390/land9020052

Thanks for pointing us to the Stepanov et al study, which is a good example to highlight in the context of this paragraph. We included it to strengthen the argument we make in our manuscript (lines 539-541).

691-692: "We conclude that LUMs and IAMs have fundamental limitations in capturing all relevant processes related to LULC changes." I don't see how your results lead to this conclusion.

There are indeed a number of features that we would argue are important limitations. For instance, none of the models investigated in this study account for gross land changes – which in turn is crucial

when assessing carbon cycle impacts ('slow in, fast out', see lines 489-494, 781-783). Additionally, these models would still underestimate the dynamics observed for example with RS products (e.g. Fuchs et al 2014 - GCB / Pongratz et al. 2017 - GCB "Model meets data" / Bayer et al 2017 - ESD), e.g. the temporal and spatial cascades of transition types consisting of multiple land use changes (forestry dynamics, shifting cultivation, agro-forestry, crop-pasture rotation systems, land abandonment, etc.). This is discussed in the first paragraph of section 4.1.

We slightly adapted the statement in the conclusion (lines 776-778) to "We conclude that LUMs and IAMs have some fundamental limitations in capturing all relevant processes related to LULC changes which in some scenarios result in questionable and potentially unrealistic features in their regional LULC allocations and their global and regional trends.". Because the primary focus of our study are not the shortcomings per se but the effects they have on future LULC patterns and ecosystem dynamics which is expressed in the next sentence. In our study we only discuss some limitations at hand. We think that the emphasis of our statement is clearer now.

---

## Author Response (AR2)

Dear Editor,

we are happy that you and the two reviewers were satisfied with our thorough revision of the manuscript. Please find attached the answers to the two open comments by reviewer 2. In addition, we applied two formal corrections for which the editorial support team asked us, i.e. adding full first author names and re-formatting Table 2 to avoid colors and reducing it in size.

Yours sincerely,
Anita Bayer et al.

**Reply to reviewer 2:**

The authors have done a great job addressing reviewers' comments. I only have two remaining minor comments:

l30-32: please add the range for these trends (as presented later in the result section). This is still missing despite my comment and is very important given that highlighting divergences/uncertainties is precisely the main point of the study.

We are sorry for this mistake and of course you are right that the uncertainty ranges are one focus point of the study. In our revision we had only added the averages to the variability ranges in the second sentence. We changed this to have both, average change and variability across scenarios, in the first sentence and no numbers in the following sentence.

L30-35: Across the diverging LULC projections we identified positive global trends of net primary productivity (+10.2% ± 1.4%), vegetation carbon (+9.2% ± 4.1%), crop production (+31.2% ± 12.2%) and water runoff (+9.3% ± 1.7%), and a negative trend of soil and litter carbon stocks (-0.5% ± 0.4%). The variability in ecosystem service indicators across scenarios was especially high for vegetation carbon stocks and crop production.

This sentence does not make sense to me grammatically: "While it is not completely impossible, of course, we argue that a speed and magnitude extremely opposing trends observed in the past seem at least questionable

We changed the sentence to (l475-477) "While it is not completely impossible, of course, we argue that a speed and magnitude which extremely oppose trends observed in the past seem at least questionable."